# Neural Circuits for Fast Poisson Compressed Sensing in the Olfactory Bulb

**Jacob A. Zavatone-Veth**[1,2,*], **Paul Masset**[1,3,*], **William L. Tong**[1,4,5], **Joseph D. Zak**[6],
**Venkatesh N. Murthy**[1,3,†], **Cengiz Pehlevan**[1,4,5,†]

[1]Center for Brain Science, [2]Department of Physics,
[3]Department of Molecular and Cellular Biology,
[4]John A. Paulson School of Engineering and Applied Sciences,
[5]Kempner Institute for the Study of Natural and Artificial Intelligence,
Harvard University
Cambridge, MA 02138
[6]Department of Biological Sciences,
University of Illinois at Chicago
Chicago, IL 60607
jzavatoneveth@g.harvard.edu, paul_masset@fas.harvard.edu,
vnmurthy@fas.harvard.edu, cpehlevan@seas.harvard.edu

## Abstract

Within a single sniff, the mammalian olfactory system can decode the identity and concentration of odorants wafted on turbulent plumes of air. Yet, it must do so given access only to the noisy, dimensionally-reduced representation of the odor world provided by olfactory receptor neurons. As a result, the olfactory system must solve a compressed sensing problem, relying on the fact that only a handful of the millions of possible odorants are present in a given scene. Inspired by this principle, past works have proposed normative compressed sensing models for olfactory decoding. However, these models have not captured the unique anatomy and physiology of the olfactory bulb, nor have they shown that sensing can be achieved within the 100-millisecond timescale of a single sniff. Here, we propose a rate-based Poisson compressed sensing circuit model for the olfactory bulb. This model maps onto the neuron classes of the olfactory bulb, and recapitulates salient features of their connectivity and physiology. For circuit sizes comparable to the human olfactory bulb, we show that this model can accurately detect tens of odors within the timescale of a single sniff. We also show that this model can perform Bayesian posterior sampling for accurate uncertainty estimation. Fast inference is possible only if the geometry of the neural code is chosen to match receptor properties, yielding a distributed neural code that is not axis-aligned to individual odor identities. Our results illustrate how normative modeling can help us map function onto specific neural circuits to generate new hypotheses.

## 1 Introduction

Sensory systems allow organisms to detect physical signals in their environments, enabling them to maximize fitness by acting adaptively. This experience of the physical environment, also known as the *Umwelt*, depends on the sensors and sensory organs of each organism [1, 2]. Throughout evolution, organisms have developed specialized sensory mechanisms to extract specific information about the

---

[*]JAZ-V and PM contributed equally to this work.
[†]VNM and CP jointly supervised this work.

37th Conference on Neural Information Processing Systems (NeurIPS 2023).

physical world. In vision and audition—the most studied sensory modalities in neuroscience—stimuli are characterized by intuitive metrics such as orientation or frequency, which have been shown to map onto neural representations from the earliest stages of the sensory systems [3–5]. One can continuously vary the orientation of an object or the pitch of a tone and quantify resulting changes in perception and neural representations. From a computational point of view, this structure in the representations can be viewed as optimizing the information transfer in the network [6–16].

In contrast, the geometric structure of the olfactory world is far less clear: How can one 'rotate' a smell? Despite significant effort, attempts to find such structure in olfactory stimuli and link that geometry to maps in olfactory areas have succeeded only in identifying coarse principles for high-level organization, far from the precision of orientation columns or tonotopy in visual and auditory cortices [17–19]. In the absence of geometric intuitions, the principles of compressed sensing (CS) have emerged as an alternative paradigm for understanding olfactory coding [20–27]. This framework provides a partial answer to the question of how an organism could identify which of millions of possible odorants are present given the activity of only a few hundred receptor types [28–33]. However, existing CS circuit models do not admit convincing biologically-plausible implementations that can perform fast inference at scale. Indeed, many proposals assume that the presence of each odorant is represented by a single, specialized neuron, which is inconsistent with the distributed odor coding observed *in vivo* [22, 34–36]. This axis-aligned coding does not leverage the geometric structure of the sensory space, which can be defined even in the absence of interpretable dimensions [37–41].

In this paper, we propose a Poisson CS model for the mammalian olfactory bulb. Our primary contributions are as follows:

- We derive a normative CS circuit model which can be mapped onto the circuits of the bulb (§3). Importantly, this mapping goes beyond basic counting of cell types; it includes detailed biological features like symmetric coupling and state-dependent inhibition (§4).

- We show that this model enables fast, accurate inference of odor identity in a biologically reasonable regime where tens of odorants are present in a given scene (§5). This fast inference is enabled by considering the geometry of the olfactory receptor code. This consideration leads to distributed odor coding, resolving a major tension between previous CS circuit models and neural data.

- We extend our circuit model to allow Bayesian inference of uncertainty in odor concentrations (§6).

In total, our results demonstrate the importance of considering representational geometry when trying to understand neural coding in the olfactory bulb (OB). Importantly, we show that it is the geometry in the space defined by the receptor affinity (or OSN activation) that controls the speed of inference. This view is distinct from previous geometric theories of olfaction, which have focused on the space of odorants [42, 43]. We propose that thinking in terms of the geometry of OSN coding will allow for deeper understanding of early olfactory processing.

## 2 Related work and review of the olfactory sensing problem

We begin with a review of the principles of olfactory coding, and of previous CS models for olfactory circuits. The structural logic of early olfactory processing is broadly conserved across the animal kingdom (Fig. 1A) [44–47], and this distinctive circuit structure is thought to play a key role in the computational function of the olfactory bulb [48–50]. In mammals, volatile odorants are first detected by olfactory receptor (ORs) proteins expressed on the surface of olfactory sensory neurons (OSNs). Each OSN expresses only a single OR type; in humans there are around 300 distinct ORs, in mice around 1000 [45]. Importantly, most ORs have broad affinity profiles, and the OSN code for odor identity is combinatorial [51]. In contrast to the immune system's highly adaptable chemical recognition capabilities arising from somatic recombination [52], ORs are hard coded into the genome as single genes [53], and therefore can only change over evolutionary timescales [54, 55]. Some adaptation of expression levels across the receptors is possible [56], but the chemical affinity of the receptor array is fixed. OSNs expressing the same OR then converge onto the same glomerulus, synapsing onto the principal projection cells of the olfactory bulb (OB), the mitral and tufted cells. These in turn send signals to olfactory cortical areas. The OB contains several types of inhibitory interneurons, whose computational role remains to be clarified [49]. Importantly, the excitatory mitral and tufted cells are not reciprocally connected across glomeruli. Instead, they are connected through a network of inhibitory granule cells, the most numerous cell type in the OB [57].

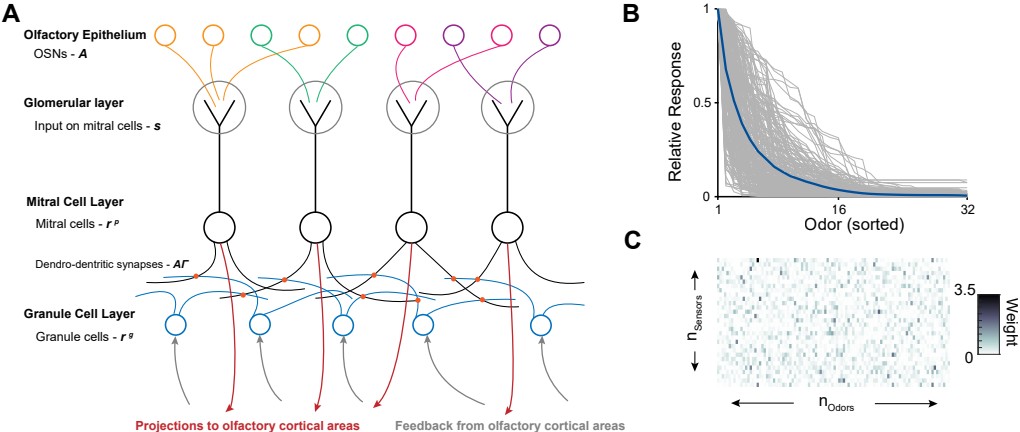

Figure 1: Circuit architecture of the mammalian olfactory bulb. **A.** Outline of the anatomy of OB circuits. **B.** Fit of the responses of 228 OSN glomerular responses to a panel of 32 odorants. (Gray line, response of single glomeruli, Blue line, Fitted average response.) See Appendix F for details of how these data were collected. **C.** Affinity matrix generated from the data-driven model ensemble. For illustrative purposes, we use only 30 receptors and 100 odorants; in simulations we use 300 receptors to roughly match humans [45], and 1000 or more odorants.

In our model, we will focus on a particular task: odor component identification within a single sniff [58, 59]. This computational problem differs from many experimental tasks focusing on discrimination between two odorants [60], which underlie most scientific work on rodent olfactory decision making [60–62]. Here, the goal is to identify the components of a complex olfactory scene [63–66]. Importantly, the limits of human performance in this setting remain to our knowledge unknown [67, 68]. To render the problem more tractable, we will make a number of simplifications of the anatomy and physiology of the OB. We will not distinguish between mitral and tufted cells, the two classes of projection neurons. Recent works have shed some light on the distinct computational roles of these cell types; these distinctions are likely to become important when considering richer environmental dynamics than we do here [69, 70]. We will also ignore the fact that odorant concentrations can vary over many orders of magnitude, and OSN responses to large changes in concentration are strongly nonlinear. Here, we will focus on concentration changes of one or two orders of magnitude. Within such ranges, the responses of OB neurons are well characterized by linear models [71].

On the theoretical side, it is widely recognized that the tremendous compression of dimensionality inherent in the transformation from odorant molecules to receptor activity means that the olfactory decoding problem is analogous to the one faced in compressed sensing (CS) [21–27, 72–74]. Classical CS theory shows that sparse high-dimensional signals can be recovered from a small number of random projections [28–33]. Inspired by these results, previous works have used the principles of CS to build circuit models for olfactory coding [21–27, 72–74]. However, these model circuits do not map cleanly onto the neural circuits of the OB and the olfactory cortical areas, particularly because they often assume each granule cell encodes exactly one odorant. Moreover, these works usually assume a Gaussian noise model for OSNs, which is biologically unrealistic (but see [21, 22]). Instead, OSN activity is better captured by a Poisson noise model [75, 76]. Some theoretical guarantees for Poisson CS are known, but the situation is less well-understood than in the Gaussian case [32, 33, 77, 78].

## 3    A neural circuit architecture for Poisson compressed sensing

We now derive a normative, rate-based neural circuit model that performs Poisson CS, which in subsequent sections we will map onto the circuitry of the OB. The design of this model will follow general biological principles, without initially drawing on specific knowledge of the OB. With the goal of sensing within a single sniff in mind, the circuit's objective is to rapidly infer the odorant concentrations $\mathbf{c} \in \mathbb{R}_+^{n_{\text{odor}}}$ underlying a single, static sample of OSN activity $\mathbf{s} \in \mathbb{R}_+^{n_{\text{OSN}}}$. For simplicity, we model the mean activity of OSNs as a linear function of the concentration, with a receptor affinity matrix $\mathbf{A} \in \mathbb{R}_+^{n_{\text{OSN}} \times n_{\text{odor}}}$ and a baseline rate $\mathbf{r}_0 \in \mathbb{R}_+^{n_{\text{OSN}}}$. As motivated above,

we use a Poisson noise model for OSN activity given the underlying concentration signal $\mathbf{c}$, and correspondingly a Gamma prior over concentrations with shape $\boldsymbol{\alpha} \in \mathbb{R}_+^{n_{\text{odor}}}$ and scale $\boldsymbol{\lambda} \in \mathbb{R}_+^{n_{\text{odor}}}$:[3]

$$\mathbf{s} \,|\, \mathbf{c} \sim \text{Poisson}(\mathbf{r}_0 + \mathbf{Ac}), \qquad \mathbf{c} \sim \text{Gamma}(\boldsymbol{\alpha}, \boldsymbol{\lambda}). \tag{1}$$

Given this likelihood and prior, we construct a neural circuit to compute the maximum *a posteriori* (MAP) estimate of the concentration $\mathbf{c}$ using gradient ascent on the log-posterior probability. Here, we sketch the derivation, deferring some details to Appendix B. Our starting point is the gradient ascent equation

$$\dot{\mathbf{c}}(t) = \boldsymbol{\nabla}_{\mathbf{c}} \log p(\mathbf{c} \,|\, \mathbf{s}) = \mathbf{A}^{\top}[\mathbf{s} \oslash (\mathbf{r}_0 + \mathbf{Ac}) - \mathbf{1}] + (\boldsymbol{\alpha} - \mathbf{1}) \oslash \mathbf{c} - \boldsymbol{\lambda}, \tag{2}$$

where $\oslash$ denotes elementwise division and $\dot{\mathbf{c}}(t) = d\mathbf{c}/dt$. Here, the estimate $\mathbf{c}$ is formally constrained to $\mathbb{R}_+^{n_{\text{odor}}}$; in numerical simulations we will sometimes ignore this constraint. Circuit algorithms of this form were studied in previous work by Grabska-Barwińska et al. [21].

However, in this most basic setup there is a one-to-one mapping between neurons and odorants, which is at variance with our knowledge of biological olfaction (§2). To distribute the code, we instead use a projected setup where the firing rates $\mathbf{g} \in \mathbb{R}^{n_{\text{g}}}$ of the neurons are mapped to the concentration estimate $\mathbf{c}$ through a matrix $\boldsymbol{\Gamma} \in \mathbb{R}^{n_{\text{odor}} \times n_{\text{g}}}$: $\mathbf{c}(t) = \boldsymbol{\Gamma}\mathbf{g}(t)$. Even if $\boldsymbol{\Gamma}$ is non-square—in particular, if $n_{\text{g}} > n_{\text{odor}}$—so long as $\boldsymbol{\Gamma}\boldsymbol{\Gamma}^{\top}$ is positive-definite and the rates follow the dynamics $\tau_{\text{g}}\dot{\mathbf{g}}(t) = (\mathbf{A}\boldsymbol{\Gamma})^{\top}[\mathbf{s} \oslash (\mathbf{r}_0 + \mathbf{A}\boldsymbol{\Gamma}\mathbf{g}) - \mathbf{1}] + \boldsymbol{\Gamma}^{\top}[(\boldsymbol{\alpha} - \mathbf{1}) \oslash (\boldsymbol{\Gamma}\mathbf{g}) - \boldsymbol{\lambda}]$ for some time constant $\tau_{\text{g}}$, the concentration estimates will still converge to the MAP. This corresponds to preconditioned gradient ascent [39, 79]. Again, we formally require the constraint that $\mathbf{c} \in \mathbb{R}_+^{n_{\text{odor}}}$, which translates into a constraint on $\mathbf{g}$. Most simply, we may take $\mathbf{g} \in \mathbb{R}_+^{n_{\text{g}}}$ and choose $\boldsymbol{\Gamma}$ to be positivity-preserving.

These dynamics include two divisive non-linearities, which can be challenging to implement in biophysical models of single neurons [80]. Using the approach proposed by Chalk et al. [81], we can linearize the inference by introducing two additional cell types that have as their fixed points the elementwise divisions $\mathbf{s} \oslash (\mathbf{r}_0 + \mathbf{A}\boldsymbol{\Gamma}\mathbf{g})$ and $(\boldsymbol{\alpha} - \mathbf{1}) \oslash (\boldsymbol{\Gamma}\mathbf{g})$. Concretely, we introduce cell types with rates $\mathbf{p} \in \mathbb{R}^{n_{\text{OSN}}}$ and $\mathbf{z} \in \mathbb{R}^{n_{\text{odor}}}$ such that their fixed-point rates for fixed $\mathbf{g}$ are $\mathbf{p}^* = \mathbf{s} \oslash (\mathbf{r}_0 + \mathbf{A}\boldsymbol{\Gamma}\mathbf{g})$ and $\mathbf{z}^* = (\boldsymbol{\alpha} - \mathbf{1}) \oslash (\boldsymbol{\Gamma}\mathbf{g})$, respectively. This yields the coupled circuit dynamics

$$\begin{aligned}
\mathbf{c}(t) &= \boldsymbol{\Gamma}\mathbf{g}(t), & \tau_{\text{g}}\dot{\mathbf{g}}(t) &= (\mathbf{A}\boldsymbol{\Gamma})^{\top}(\mathbf{p} - \mathbf{1}) + \boldsymbol{\Gamma}^{\top}(\mathbf{z} - \boldsymbol{\lambda}), \\
\tau_{\text{p}}\dot{\mathbf{p}}(t) &= \mathbf{s} - \mathbf{p} \odot (\mathbf{r}_0 + \mathbf{A}\boldsymbol{\Gamma}\mathbf{g}), & \tau_{\text{z}}\dot{\mathbf{z}}(t) &= \boldsymbol{\alpha} - \mathbf{1} - \mathbf{z} \odot \mathbf{c},
\end{aligned} \tag{3}$$

for cell-type-specific time constants $\tau_{\text{g}}$, $\tau_{\text{p}}$, and $\tau_{\text{z}}$, where $\odot$ denotes elementwise multiplication. In the limit $\tau_{\text{p}}, \tau_{\text{z}} \ll \tau_{\text{g}}$, this circuit will recover the MAP gradient ascent. If $\tau_{\text{p}}$ and $\tau_{\text{z}}$ are not infinitely fast relative to $\tau_{\text{g}}$, we expect these dynamics to approximate the desired dynamics [81] (see Appendix C for a preliminary analysis of the linear stability of the MAP fixed-point). We will test the accuracy of this approximation for biologically-reasonable time constants using numerical experiments. Moreover, $\mathbf{p}$ should formally be constrained to $\mathbb{R}_+^{n_{\text{p}}}$, such that the non-negativity of the target ratio is respected. Finally, we note that in the special case $\boldsymbol{\alpha} = \mathbf{1}$ in which the Gamma prior reduces to an exponential prior, the introduction of the cell type $\mathbf{z}$ is no longer required.

## 4 Biological interpretation and predictions of the circuit model

We now argue that the normative model derived in the preceding section can be mapped onto the circuitry of the OB. In particular, though the model was derived based only on general biological principles, its specific features are biologically implementable based on the detailed anatomy and physiology of the OB. In terms of the levels of understanding of neural circuits proposed by David Marr, this is an example of how normative modeling can bridge the gap between algorithmic and mechanistic understanding [82, 83].

As foreshadowed by our notation, we interpret the cell type $\mathbf{g}$ as the granule cells of the OB, and the cell type $\mathbf{p}$ as the mitral cells, which are projection neurons. Provided that the elements of the matrix $\mathbf{A}\boldsymbol{\Gamma}$ are non-negative, this interpretation is justified at the coarsest level by the signs with which the two cell types appear in the dynamics: the $\mathbf{p}$ neurons excite the $\mathbf{g}$ neurons, which in turn inhibit the $\mathbf{p}$ neurons. Finally, we interpret the cell type $\mathbf{z}$ as representing a form of cortical feedback. In the remainder of this section, we will justify this mapping in detail.

---

[3]See Appendix A for a detailed description of our notational conventions.

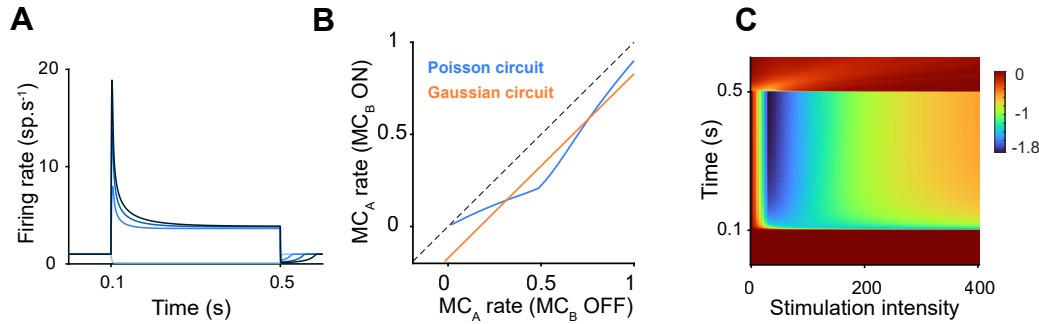

Figure 2: State-dependent inhibition of mitral cells. **A.** Upon stimulation, mitral cells exhibit a transient burst of activity followed by relaxation to a plateau. Darker color indicates stronger stimulation. **B.** In the circuit for Poisson CS we propose (blue line), the inhibition due to the activation of a second mitral cell ($MC_B$) is gated by the activity of the cell we are recording from ($MC_A$), as observed experimentally by Arevian et al. [95] in their Fig. 2D. This state-dependent gating does not occur in a circuit derived from a Gaussian noise model (orange line). Rates are normalized to the maximal stimulation of the principal cell. Dashed line indicates the unity line. **C.** Strength of inhibition as a function of time and of stimulation intensity; c.f. [95] Fig. 3B.

### 4.1 Circuit anatomy: weight transport, dendro-dendritic coupling, and cortical feedback

The first salient feature of the dynamics (3) is that the cell types **g** and **p** do not make direct lateral connections amongst themselves. Rather, they connect only indirectly through the neurons of the opposing cell type, matching the connectivity structure of mitral and granule cells (Fig. 1A). Moreover, the synaptic weights $\mathbf{A\Gamma}$ of the connections from **g** to **p** neurons mirror exactly the weights $(\mathbf{A\Gamma})^\top$ of the connections from **p** to **g** neurons. Naïvely, this creates a weight transport problem of the form that renders backpropagation biologically implausible: how should the exact transpose of a matrix be copied to another synapse [84–86]? However, in the unique case of the OB this does not pose a substantial obstacle, as the mitral and granule cells are coupled by dendro-dendritic synapses, meaning that bi-directional connectivity occurs at a single physical locus (Fig. 1) [49, 57, 87].

This interpretation accounts for the interactions between the cell types **g** and **p** in the dynamics (3), but what about the cell type **z**? These cells give direct excitatory input to the granule cells **g** with weights $\mathbf{\Gamma}^\top$, and represent the concentration-dependent contribution to the log-prior gradient. We can therefore interpret these cells as representing feedback from olfactory cortical areas to the OB, which arrives at the granule cells [88–90]. Though our model can thus flexibly incorporate cortical feedback, for our subsequent simulations we will focus on the simple case in which the prior is static and has $\boldsymbol{\alpha} = \mathbf{1}$, in which case it reduces to an exponential and feedback is not explicitly required.

Given this mapping of the cell types of our model to the cell types of the bulb, we will henceforth choose the membrane time constants to match experiment, taking $\tau_p = 20$ ms [91] and $\tau_g = 30$ ms [92]. This matching of timescales is required for our comparison of model inference timescales to the timescale of a single sniff to be meaningful.

### 4.2 Choosing the affinity matrix

In theories of Poisson CS, the optimal sensing matrix **A** is one that has columns that are in some precise sense as orthogonal as possible, so that it acts as an approximate isometry [27–33]. However, biologically, the olfactory system is not free to choose optimal sensors. Rather, the affinity profiles of each receptor are dictated by biophysics and by evolutionary history [53, 54, 93]. To build a more realistic model for OSN sensing, we therefore turn to biological data. Using two-photon calcium imaging, we recorded the responses of 228 mouse OSNs glomeruli to 32 odorants (see Appendix F for details; these data were previously published in [94]). In Fig. 1B, we show that the distribution of responses is well-fit by a Gamma distribution with shape 0.37 and scale 0.36 (Appendix F). We then define our ensemble of sensing matrices **A** by drawing their elements as independently and identically distributed $\mathrm{Gamma}(0.37, 0.36)$ random variables. An example matrix drawn from this ensemble is shown in Fig. 1C.

### 4.3 Divisive predictive coding by mitral cells

In our model (3), odorant concentration estimates are decoded from granule cell activity. This feature is shared with previous Gaussian CS models of olfactory coding [20, 24, 25]. Our model, however, matches biology more closely than these previous works because it allows for a distributed code rather than assuming that each granule cell codes for a single odorant.

What, then, is the functional role of the mitral cells in our model? We can interpret their dynamics as implementing a form of predictive coding, in which they are trying to cancel their input by the current prediction. Because the mitral cell activity converges to the ratio of their input to the prediction, this a divisive form of predictive coding [81]. In contrast, a Gaussian noise model gives a subtractive form of predictive coding in which the activity converges to the difference between input and prediction (Appendix D) [15]. In Fig. 2A, we show example timecourses of model mitral cell activity following the onset of a stimulus. Consistent with experimentally-measured responses [96, 97], a sharp transient response at the onset of stimulation is followed by decay to a low level of tonic activity (Fig. 2A).

### 4.4 State-dependent inhibition of mitral cells

A salient feature of our circuit model is that the inhibition from the granule cells onto a mitral cell is gated by the activity of the mitral cell itself (3). This state-dependent inhibition is reminiscent of *in vitro* experiments showing that granule cell mediated inhibition is activity dependent [95]. In these experiments, Arevian et al. measured the activity of a primary mitral cell ($MC_A$) while increasing its level of stimulation under two conditions. In the first condition, no other cells in the circuit are being stimulated. In the second condition, they also activate another mitral cell ($MC_B$). The activation of the second mitral cell leads to the activation of granule cells connected to both mitral cells and a reduction in the firing evoked by stimulation of the primary cell alone. Strikingly, these authors showed that this inhibition is dependent on the activity of the primary cell [95].

Here, we show that our proposed circuit reproduces these observations of state-dependent inhibition, and that they do not arise in a similarly-constructed circuit for a Gaussian noise model. To model Arevian et al. [95]'s *in vitro* experiments, we simulated a reduced circuit with 2 mitral cells and 10 granule cells. We stimulated $MC_A$ with an input $s_A \in [1 : 400]$ while toggling on or off the stimulation $s_B = 80$ of the second mitral cell $MC_B$. As observed experimentally, when $MC_B$ is activated, the inhibition of the primary mitral cell $MC_A$ is state dependent (Fig. 2B). To show that this effect arises from our Poisson circuit, we build a similar inference circuit with a Gaussian noise model (Appendix D). Under similar conditions, the inhibition in the Gaussian circuit is independent of the activity of the primary mitral cell $MC_A$ (Fig. 2B). Furthermore, the dynamics of the relative inhibition qualitatively recapitulate those observed experimentally, with sustained inhibition throughout the stimulation period at the stimulation level of $MC_A$ leading to maximal inhibition and inhibition followed by relaxation for stronger stimulation levels of $MC_A$ (Fig. 2C and see [95] Fig. 3b).

## 5 Geometry, speed, and capacity

We have argued that the model introduced in §3 could be implemented biologically, but can it perform at scale? Concretely, can a circuit of this architecture with neuron counts comparable to the human OB correctly identify which among a large set of odorants are present in a given scene? This is precisely the question of the capacity of the CS algorithm [98, 99]. In Fig. 3A, we present our algorithm with scenes composed of varying numbers of randomly-selected odorants out of a panel of 1000, which for simplicity we take to be at the same concentration. We first ask how many odorants can be reliably detected within a single sniff, i.e., 200 ms. To convert MAP concentration estimates into presence estimates, we simply binarize the estimated concentrations based on whether they are larger than half of the true odorant concentration. The ability of the one-to-one code to successfully detect odorants falls off rapidly, with the detection fraction falling below one-half even if only a handful of odorants are present (Fig. 3A).[4]

The limited capacity of the one-to-one code can be overcome by distributing the code in a way that takes into account the geometry of the sensing problem. Here, the information geometry of the

---

[4]We remark that, with a one-to-one code, our model is identical to that proposed by Grabska-Barwińska et al. [21] except for the introduction of the granule cells.

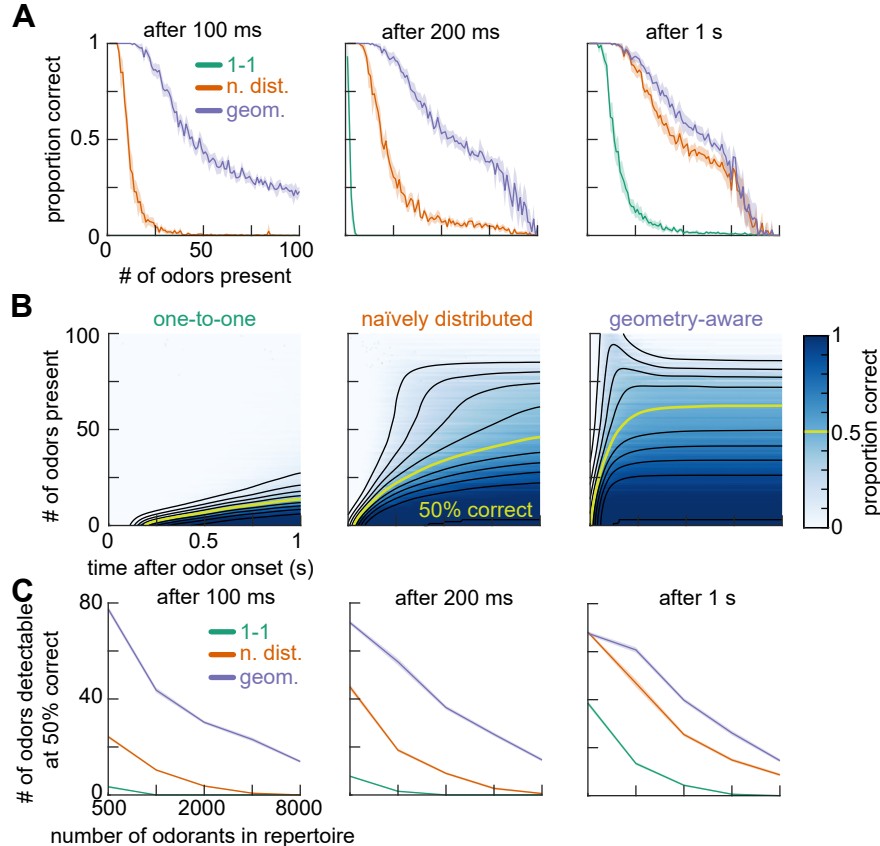

Figure 3: Fast detection of many odorants. **A**. Fraction of odorants correctly detected within 100 ms (*left*), 200 ms (*center*), and 1 s (*right*) after odorant onset as a function of the number of odorants present, for models with one-to-one, naïvely distributed, and geometry-aware codes. Here, we consider a repertoire of 1000 possible odorants. **B**. Heatmap with overlaid smoothed contours of correct detection fraction as a function of number of present odorants and time window for models with one-to-one (*left*), naïvely distributed (*center*), and geometry-aware (*right*) codes. As in **A**, a repertoire of 1000 possible odorants is used. **C**. Threshold number of odorants for which half can be correctly detected as a function of total repertoire size within 100 ms (*left*), 200 ms (*center*), and 1 s (*right*) after odorant onset. See also Supp. Figs. G.1 and G.2 for versions of panels **A** and **B** with varying repertoire sizes, from which the capacities shown here are derived. See Appendix G for details of our numerical methods. Shaded patches show $\pm 1.96$ SEM over realizations throughout.

problem is governed by the sensing matrix $\mathbf{A}$, which introduces correlations in the input signals to mitral cells because the off-diagonal components of $\mathbf{A}^\top \mathbf{A}$ are non-negligible. To counteract these detrimental correlations, we can choose a readout matrix $\boldsymbol{\Gamma}$ such that $\boldsymbol{\Gamma}\boldsymbol{\Gamma}^\top \approx (\mathbf{A}^\top \mathbf{A})^+$ up to constants of proportionality, where we must take a pseudoinverse because $\mathbf{A}^\top \mathbf{A}$ is highly rank-deficient (see Appendix G for details). Geometrically, this corresponds an approximation of natural gradient descent, in which we use the Gauss-Newton matrix instead of the Fisher information matrix because the latter is state-dependent for Poisson likelihoods [37–40]. As a control, we also consider a naïvely distributed code with $\boldsymbol{\Gamma}\boldsymbol{\Gamma}^\top \approx \mathbf{I}_{n_{\text{odor}}}$. Distributing the code, even without accounting for the geometry of inference, markedly improves the single-sniff capacity to around 10-20, and taking into account the geometry produces a further improvement (Fig. 3A).

To gain a more granular view of how tuned geometry enables faster inference, in Fig. 3B we test the three models' detection capabilities at sub-sniff resolution. We can see that at long times—around a second after odorant onset—the naïvely distributed and geometry-aware codes achieve similar capacities of around 50-60 odorants. However, the geometry-aware code reaches this detection capacity within a single sniff, whereas the naïvely distributed code requires the full second of

processing time. The detection capacity of the one-to-one code reaches only around 20 odorants after 1 second, and even then does not appear to have reached its asymptote. These results illustrate two important conceptual points: First, when the strength of individual synapses is bounded, distributed coding can speed up dynamics by increasing the effective total input to a given neuron. Second, given a sensing matrix that induces strong correlations, geometry-aware distributed coding can accelerate inference by counteracting that detrimental coupling.

These tests show show that our model can reliably detect tens of odorants from a panel of thousands of possible odorants, but do not probe how the model's capacity scales to larger odor spaces. While the true dimensionality of odor space remains unknown [100, 101], there may be orders of magnitude more than thousands of possible odorants. As an upper bound, there are on the order of millions of known volatile compounds that are plausibly odorous [102]. Thus, it is important to determine how our model scales to more realistically-sized odor spaces. From the literature on compressed sensing performance bounds, we expect the threshold sparsity to decay slowly—roughly logarithmically— with increasing $n_{\mathrm{odor}}$ [28–33]. As a first step, in Supp. Figs. G.1 and G.2, we reproduce Fig. 3A-B for between 500 and 8000 possible odorants, showing that performance does indeed drop off slowly with increasing odor space dimension. To get a more precise estimate of how performance scales with repertoirse size, in Fig. 3C we plot the threshold number of odorants for which half can be reliably detected as a function of the repertoire size, showing that for the geometry-aware code it decays only a bit faster than logarithmically. Our ability to simulate larger systems was limited by computational resources. This limitation is present in previous works, and the repertoires tested here are comparable to—or substantially larger than—those used in past studies [21, 22, 24, 26, 72].

## 6    Fast sampling for uncertainty estimation

Thus far, we have focused on a circuit that performs MAP estimation of odorant concentrations. However, to successfully navigate a dynamic, noisy world, animals must estimate sensory uncertainty at the timescale of perception [38, 103–107]. Fortunately, our circuit model can be easily extended to perform Langevin sampling of the full posterior distribution, allowing for uncertainty estimation while maintaining its attractive structural features. In Appendix B, we provide a detailed derivation of a model that implements Langevin sampling through the granule cells for a Poisson likelihood and Gamma prior. This yields a circuit that is identical to the MAP estimation circuit introduced in §3 up to the addition of Gaussian noise to the granule cell dynamics:

$$\mathbf{c}(t) = \mathbf{\Gamma}\mathbf{g}(t) \qquad \tau_{\mathrm{g}}\dot{\mathbf{g}}(t) = (\mathbf{A}\mathbf{\Gamma})^{\top}(\mathbf{p} - \mathbf{1}) + \mathbf{\Gamma}^{\top}(\mathbf{z} - \boldsymbol{\lambda}) + \boldsymbol{\xi}(t)$$
$$\tau_{\mathrm{p}}\dot{\mathbf{p}}(t) = \mathbf{s} - \mathbf{p} \odot (\mathbf{r}_0 + \mathbf{A}\mathbf{\Gamma}\mathbf{g}) \qquad \tau_{\mathrm{z}}\dot{\mathbf{z}}(t) = \boldsymbol{\alpha} - \mathbf{1} - \mathbf{z} \odot \mathbf{c}. \tag{4}$$

Here, $\boldsymbol{\xi}(t)$ is a vector of $n_{\mathrm{g}}$ independent zero-mean Gaussian noise processes with covariance $\mathbb{E}[\xi_j(t)\xi_{j'}(t')] = 2\tau_{\mathrm{g}}\delta_{jj'}\delta(t - t')$, and once again the rates $\mathbf{g}$ and $\mathbf{p}$ should in principle be constrained to be non-negative. In this case, the readout matrix $\mathbf{\Gamma}$ both preconditions the effective gradient force and shapes the structure of the effective noise $\mathbf{\Gamma}\boldsymbol{\xi}$, allowing us to mold the geometry of the sampling manifold [37, 40]. By using a projected readout, we maintain the independence of the noise processes for different neurons. This both allows us to have many independent samplers—if $n_{\mathrm{g}} > n_{\mathrm{odor}}$—and is important for biological realism if we interpret the sampling noise as resulting from fluctuations in membrane potential due to synaptic noise [108].

As a test of how this sampling circuit performs, we consider a simple setup in which one set of odorants appears at a low concentration, and then a second set of odorants appear at a higher concentration while the low odorants are still present. This setup tests both the circuit's ability to converge rapidly enough to give accurate posterior samples within a single sniff, and its ability to correctly infer odorant concentrations even in the presence of distractors [63, 66]. In Fig. 4, we show that circuits with one-to-one or naïvely distributed codes do not give accurate estimates of the concentration mean within 200 ms, while tuning the geometry to match the receptor affinities enables fast convergence. All three circuits overestimate the posterior variance at short times, consistent with what one would expect for unadjusted Langevin samplers [109], but the geometry-aware model's estimate decays most rapidly towards the target. Therefore, when the synaptic weights are tuned, our circuit model can enable fast, robust estimation of concentration statistics, even in the presence of distractors.

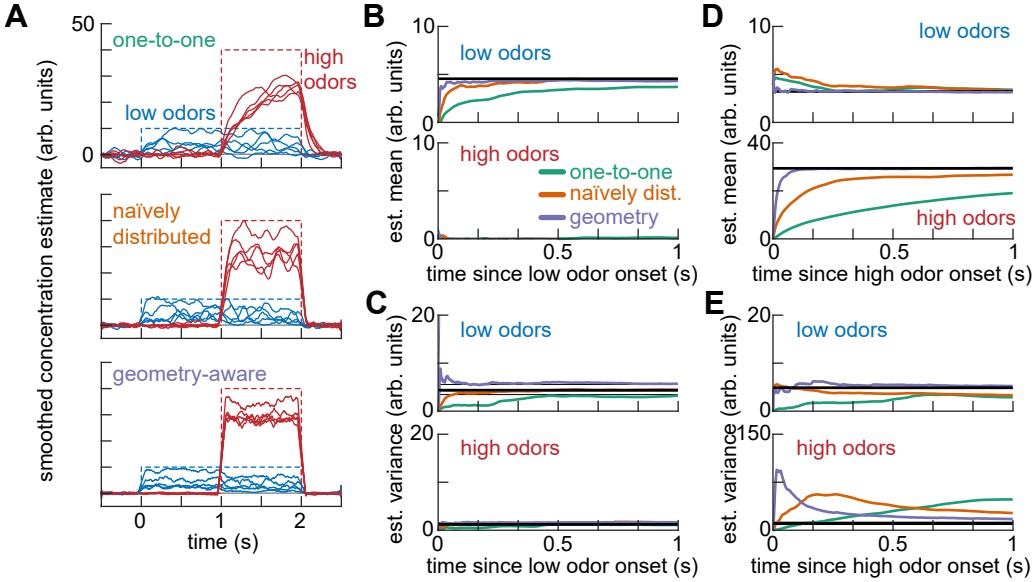

Figure 4: Fast uncertainty estimation using Langevin sampling of the posterior. Here, we use a simple concentration estimation task in which 5 randomly-selected 'low' odorants out of a panel of 1000 appear at concentration 10 at time 0 s, and then a further 5 randomly-selected 'high' odorants appear at concentration 40 at time 1 s. **A**. Smoothed timeseries of instantaneous concentration estimates for low, high, and background odorants, for models with one-to-one (*top*), naïvely distributed (*middle*), and geometry-aware (*bottom*) codes. Background odorant estimates are shown as mean ± standard deviation over odorants. Dashed lines show true concentrations over time. **B**. Cumulative estimates of concentration mean for low (*top*) and high (*bottom*) odorants after the onset of the low odorants for one-to-one, naïvely distributed, and geometry-aware codes. Black lines indicate baseline estimates of the posterior mean. Thick colored lines indicate means over odorants. **C**. As in **B**, but for the estimated variance. **D**. As in **B**, but after the onset of high odorants. **E**. As in **C**, but after the onset of high odorants. See Appendix G for details of our numerical methods, and for individual-odor traces.

## 7 Discussion

In this paper, we have derived a novel, minimal rate model for fast Bayesian inference in early olfaction. Unlike previously-proposed algorithms for CS in olfaction, this model has a clear mapping onto the circuits of the mammalian OB. We showed that this model successfully performs odorant identification across a biologically-relevant range of scene sparsities and circuit sizes. This model therefore exemplifies how normative approaches can blur the lines between algorithmic and mechanistic understanding of neural circuits [82, 83]. We now conclude by discussing possible avenues for future inquiry, as well as some of the limitations of our work.

One limitation of our simulations is that we have chosen to distribute the neural code randomly, and have allowed for negative entries in the mitral-granule synaptic weight matrix $\mathbf{A\Gamma}$ (see methods in Appendix G). These features are not entirely biologically satisfactory. However, our model only captures the mitral cells and granule cells, overlooking a number of other inhibitory cell types that could contribute to solving this problem, e.g., through feedforward inhibition onto mitral cells or lateral inhibition across granule cells [48, 49]. Fundamentally, negative values in the geometry-aware decoding matrix $\mathbf{\Gamma}$ arise as although the affinity matrix is positive, its inverse will contain negative elements. A biologically-plausible realization of the geometry-aware code through the introduction of additional cell types could be achieved by decomposing the inverse into several components, yielding sparse, consistently-signed connectivity [110]. As a first step, in Supp. Fig. G.4 we show that similar performance to Fig. 4 can be achieved using a sparse non-negative randomly distributed code. As a result, one objective for future work will be to develop better models for the decoding matrix $\mathbf{\Gamma}$ that result in more realistic connectivity.

Though our model captures two of the interesting features of the anatomy and physiology of the OB—symmetric dendrodendritic coupling and state-dependent inhibition of mitral cells—there are

many biological details which we have not addressed. First, our linear model for OSN mean firing neglects receptor antagonism, gain control, and other nonlinear effects [94, 111, 112], which are known to affect the performance of Gaussian CS models for olfaction [26, 113]. Second, our models are rate-based, while neurons in the OB spike. In spiking implementations of sampling networks, the noise is not uncorrelated across neurons, complicating their biological interpretation [38, 114]. Constructing models that capture these richly nonlinear effects will be an important objective for future work. A first step towards such a nonlinear model would be to build a spiking network that approximates the rate-based models considered here, which could be accomplished using the efficient balanced network formalism for distributed spiking networks [38]. Another step would be to add a Hill function nonlinearity to the OSN model to approximate competitive binding, as studied for Gaussian compressed sensing by Qin et al. [26]. One challenge in constructing models that incorporate additional nonlinearity is that the simple linear strategy for distributing the code used here may no longer be directly applicable. In Appendix E, we illustrate this obstacle for the relatively simple case of a model with the same linear OSNs and Poisson likelihood but an $L_0$ instead of Gamma prior, building on recent work on circuits for Gaussian CS with $L_0$ priors [115].

A closely related point is that we model the weights of the synapses between mitral and granule cells as fixed, and do not consider synaptic plasticity. In particular, we assume that they are tuned to the statistics of the receptor affinities without specifying a mechanism by which this tuning could take place. In biology, receptor abundances and other OSN properties display activity-dependent adaptation over long timescales, meaning that the optimal tuning is unlikely to be static [56, 116]. Some past works have sought to incorporate plasticity of the mitral-granule cell synapses into decoding models [24, 117], tying into a larger body of research on how plasticity can enable flexible feature extraction in olfaction [118, 119]. This learning should lead to measurable changes in the population response to odorant panels, which our model predicts should be linked in a precise way to account for receptor-induced correlations in the responses to the most frequently present odorants. Experimentally characterizing and carefully modeling these changes in responses across timescales will be an interesting avenue for future work [120]. Experimental techniques to probe these ideas at the neural and behavioral level have recently been proposed [121, 122] which allow more precise control of stimulus and subjective geometry.

Though the circuit model derived in §3 incorporates a general Gamma prior represented by cortical feedback, our simulations focus on the special case in which the prior reduces to an exponential, in which the feedback neurons are not needed. Future work will therefore be required to carefully probe the effect of incorporating a Gamma prior with non-unit shape and to dissect the structure of the resulting modeled cortical feedback. More generally, it will be interesting to extend our framework to incorporate data-adaptive priors. Importantly, the stimuli used in this paper constitute an extremely impoverished model for the richness of the true odor world; we do not account for its rich dynamical structure and co-occurence statistics. Adaptive priors as encoded by cortical feedback would allow circuits to leverage this structure, enabling faster and more accurate inference [88–90, 123, 124].

We conclude by noting that our work provides an example of how distributed coding can lead to faster inference than axis-aligned disentangled coding. In recent years, the question of when axis-aligned coding is optimal has attracted significant attention in machine learning and neuroscience [125–135]. Much of this work focuses on the question of when axis-aligned codes are optimal for energy efficiency or for generalization, whereas here we focus on the question of which code yields the fastest inference dynamics. These ideas are one example of the broader question of how agents and algorithms should leverage the rich geometry of the natural world to enable fast, robust learning and inference [136]. We believe that investigating how task demands and biological constraints affect the optimal representational geometry for that task is a promising avenue for illuminating neural information processing in brains and machines [41, 136, 137].

## Acknowledgments and Disclosure of Funding

We thank Naoki Hiratani, Shanshan Qin, and Vikrant Kapoor for useful discussions. This work was supported by NSF grants DMS-2134157 and CAREER IIS-2239780 to CP, NIH grants R01DC017311 and R01DC016289 to VNM, and NTT Research award A47994 to VNM. CP received additional support from a Sloan Research Fellowship. PM was partially supported by a grant from the Harvard Mind Brain Behavior Interfaculty Initiative. JDZ was partially supported by NIH grant K99DC017754. This work has been made possible in part by a gift from the Chan Zuckerberg Initiative Foundation to establish the Kempner Institute for the Study of Natural and Artificial Intelligence. A subset of the computations in this paper were run on the FASRC cluster supported by the FAS Division of Science Research Computing Group at Harvard University.

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

## A  Notational conventions

In this Appendix, we define the notational conventions used throughout the paper. We write $\mathbb{R}^n_+ = \{\mathbf{x} \in \mathbb{R}^n : x_1, \ldots, x_n \geq 0\}$ for the non-negative reals, and $\mathbb{N} = \{0, 1, \ldots\}$ for the natural numbers. For a vector $\boldsymbol{\lambda} \in \mathbb{R}^n_+$, we write

$$\mathbf{x} \sim \mathrm{Poisson}(\boldsymbol{\lambda}) \tag{A.1}$$

if the probability mass function of $\mathbf{x} \in \mathbb{N}^n$ is

$$p(\mathbf{x}) = \prod_{j=1}^{n} \frac{\lambda_j^{x_j}}{x_j!} e^{-\lambda_j}. \tag{A.2}$$

For scalars $\alpha > 0$ and $\lambda > 0$, we write that a vector $\mathbf{x} \in \mathbb{R}^n_+$ is distributed as

$$\mathbf{x} \sim \mathrm{Gamma}(\alpha, \lambda) \tag{A.3}$$

if its probability density function is

$$p(\mathbf{x}) = \prod_{j=1}^{n} \frac{x_j^{\alpha-1}}{\Gamma(\alpha)} e^{-\lambda x_j}. \tag{A.4}$$

Similarly, for vectors $\boldsymbol{\alpha}, \boldsymbol{\lambda} \in \mathbb{R}^n_+$, we write

$$\mathbf{x} \sim \mathrm{Gamma}(\boldsymbol{\alpha}, \boldsymbol{\lambda}) \tag{A.5}$$

if

$$p(\mathbf{x}) = \prod_{j=1}^{n} \frac{x_j^{\alpha_j-1}}{\Gamma(\alpha_j)} e^{-\lambda_j x_j}. \tag{A.6}$$

For vectors $\mathbf{x}, \mathbf{y} \in \mathbb{R}^n_+$, we write $\mathbf{x} \odot \mathbf{y}$ for their Hadamard (elementwise) product

$$(\mathbf{x} \odot \mathbf{y})_i = x_i y_i \tag{A.7}$$

and $\mathbf{x} \oslash \mathbf{y}$ for their elementwise ratio

$$(\mathbf{x} \oslash \mathbf{y})_i = \frac{x_i}{y_i}. \tag{A.8}$$

## B  Detailed derivation of circuit model

In this Appendix, we give a detailed derivation of the circuit models introduced in Sections 3 and 6 of the main text. Here, we focus on the setting of the sampling circuit; the circuit algorithm to compute the MAP introduced in Section 3 of the main text can be recovered at each step by dropping the additive noise terms from the dynamics.

We recall that our goal is to sample the posterior $p(\mathbf{c} \,|\, \mathbf{s})$ over concentrations $\mathbf{c}$ given OSN activity $\mathbf{s}$, for a Poisson likelihood

$$\mathbf{s} \,|\, \mathbf{c} \sim \mathrm{Poisson}(\mathbf{r}_0 + \mathbf{A}\mathbf{c}) \tag{B.1}$$

and a Gamma prior

$$\mathbf{c} \sim \mathrm{Gamma}(\boldsymbol{\alpha}, \boldsymbol{\lambda}). \tag{B.2}$$

Here, we use the conventions of Appendix A for Poisson and Gamma random vectors. Discarding normalization constants that do not depend on $\mathbf{c}$, the density of the posterior is then

$$p(\mathbf{c} \,|\, \mathbf{s}) \propto p(\mathbf{s} \,|\, \mathbf{c}) p(\mathbf{c}) \tag{B.3}$$

$$\propto \left[ \prod_{k=1}^{n_{\mathrm{OSN}}} (\mathbf{r}_0 + \mathbf{A}\mathbf{c})_k^{s_k} e^{-(\mathbf{r}_0 + \mathbf{A}\mathbf{c})_k} \right] \left[ \prod_{j=1}^{n_{\mathrm{odor}}} c_j^{\alpha_j-1} e^{-\lambda_j c_j} \right]. \tag{B.4}$$

We will sample from this distribution using Langevin dynamics, without explicitly constraining the concentration estimate to be non-negative [21, 22]. This corresponds to the stochastic dynamics

$$\dot{\mathbf{c}}(t) = \boldsymbol{\nabla}_{\mathbf{c}} \log p(\mathbf{c} \,|\, \mathbf{s}) + \boldsymbol{\eta}(t), \tag{B.5}$$

where $\boldsymbol{\eta}(t)$ is an $n_{\mathrm{odor}}$-dimensional zero-mean Gaussian noise process with $\mathbf{E}[\eta_j(t)\eta_{j'}(t')] = 2\delta_{jj'}\delta(t - t')$. The gradient of the log-posterior is

$$\frac{\partial \log p(\mathbf{c} \,|\, \mathbf{s})}{\partial c_j} = \sum_{k=1}^{n_{\mathrm{OSN}}} \left( \frac{s_k}{(\mathbf{r}_0 + \mathbf{Ac})_k} - 1 \right) A_{kj} + \frac{\alpha_j - 1}{c_j} - \lambda_j, \tag{B.6}$$

or, in vector notation,

$$\boldsymbol{\nabla}_{\mathbf{c}} \log p(\mathbf{c} \,|\, \mathbf{s}) = \mathbf{A}^\top [\mathbf{s} \oslash (\mathbf{r}_0 + \mathbf{Ac}) - \mathbf{1}] + (\boldsymbol{\alpha} - \mathbf{1}) \oslash \mathbf{c} - \boldsymbol{\lambda}. \tag{B.7}$$

The Langevin dynamics then become

$$\dot{\mathbf{c}}(t) = \mathbf{A}^\top [\mathbf{s} \oslash (\mathbf{r}_0 + \mathbf{Ac}) - \mathbf{1}] + (\boldsymbol{\alpha} - \mathbf{1}) \oslash \mathbf{c} - \boldsymbol{\lambda} + \boldsymbol{\eta}(t). \tag{B.8}$$

This is closely related to the Langevin sampling algorithm introduced in equation (3.9) of Grabska-Barwińska et al. [21], but here we are not separately inferring odor concentration and odor presence. We note that, for $\alpha_j > 1$, the term $(\boldsymbol{\alpha} - \mathbf{1}) \oslash \mathbf{c}$ gives a force that diverges for small $c_j$.

However, in this setup single neurons encode single odors. We instead want to allow for a distributed code, where the population of neurons responsible for sampling may not directly encode single odors. To distribute the code, we follow previous work by Masset et al. [38] in using the "complete recipe" for stochastic gradient MCMC [37]. From the "complete recipe", we know that the dynamics

$$\tau_{\mathrm{g}}\dot{\mathbf{c}}(t) = \boldsymbol{\Gamma}\boldsymbol{\Gamma}^\top \boldsymbol{\nabla}_{\mathbf{c}} \log p(\mathbf{c} \,|\, \mathbf{s}) + \boldsymbol{\Gamma}\boldsymbol{\xi}(t), \tag{B.9}$$

for a (expectantly named) time constant $\tau_{\mathrm{g}} > 0$ and any (potentially non-square) matrix $\boldsymbol{\Gamma} \in \mathbb{R}^{n_{\mathrm{odor}} \times n_{\mathrm{g}}}$ such that $\boldsymbol{\Gamma}\boldsymbol{\Gamma}^\top$ is positive-definite, will have as their stationary distribution the posterior $p(\mathbf{c} \,|\, \mathbf{s})$. here, we have replaced the $n_{\mathrm{odor}}$-dimensional noise process $\boldsymbol{\eta}(t)$ with an $n_{\mathrm{g}}$-dimensional noise process $\boldsymbol{\xi}(t)$ with zero mean and covariance

$$\mathbb{E}[\xi_j(t)\xi_{j'}(t')] = 2\tau_{\mathrm{g}}\delta_{jj'}\delta(t - t'). \tag{B.10}$$

Then, we can write the concentration estimate as

$$\mathbf{c}(t) = \boldsymbol{\Gamma}\mathbf{g}(t) \tag{B.11}$$

where the activity of the neurons $\mathbf{g} \in \mathbb{R}^{n_{\mathrm{g}}}$ follows the dynamics

$$\tau_{\mathrm{g}}\dot{\mathbf{g}}(t) = \boldsymbol{\Gamma}^\top \boldsymbol{\nabla}_{\mathbf{c}} \log p(\mathbf{g} \,|\, \mathbf{s})\Big|_{\mathbf{c}=\boldsymbol{\Gamma}\mathbf{g}} + \boldsymbol{\xi}(t). \tag{B.12}$$

Using the gradient of the log-posterior computed above, we then have

$$\tau_{\mathrm{g}}\dot{\mathbf{g}}(t) = (\mathbf{A}\boldsymbol{\Gamma})^\top [\mathbf{s} \oslash (\mathbf{r}_0 + \mathbf{A}\boldsymbol{\Gamma}\mathbf{g}) - \mathbf{1}] + \boldsymbol{\Gamma}^\top [(\boldsymbol{\alpha} - \mathbf{1}) \oslash (\boldsymbol{\Gamma}\mathbf{g})] - \boldsymbol{\Gamma}^\top \boldsymbol{\lambda} + \boldsymbol{\xi}(t). \tag{B.13}$$

These dynamics include two divisive non-linearities, which can be complex to implement in biophysical models of single neurons [80]. Using the approach proposed in Chalk et al. [81], we can linearize the inference by introducing two additional cell types that have as their fixed points the elementwise divisions $\mathbf{s} \oslash (\mathbf{r}_0 + \mathbf{A}\boldsymbol{\Gamma}\mathbf{g})$ and $(\boldsymbol{\alpha} - \mathbf{1}) \oslash (\boldsymbol{\Gamma}\mathbf{g})$. We then introduce a population $\mathbf{p}$ of $n_{\mathrm{OSN}}$ neurons, with dynamics

$$\tau_{\mathrm{p}}\dot{\mathbf{p}}(t) = \mathbf{s} - \mathbf{p} \odot (\mathbf{r}_0 + \mathbf{A}\boldsymbol{\Gamma}\mathbf{g}), \tag{B.14}$$

such that the fixed point is

$$\mathbf{p}^* = \mathbf{s} \oslash (\mathbf{r}_0 + \mathbf{A}\boldsymbol{\Gamma}\mathbf{g}). \tag{B.15}$$

We finally introduce a third population $\mathbf{z}$ of $n_{\mathrm{odor}}$ neurons, with dynamics

$$\tau_{\mathrm{z}}\dot{\mathbf{z}}(t) = \boldsymbol{\alpha} - \mathbf{1} - \mathbf{z} \odot (\boldsymbol{\Gamma}\mathbf{g}), \tag{B.16}$$

such that the fixed point is

$$\mathbf{z}^* = (\boldsymbol{\alpha} - \mathbf{1}) \oslash (\boldsymbol{\Gamma}\mathbf{g}). \tag{B.17}$$

To be more precise, the cell types $\mathbf{p}$ and $\mathbf{z}$ compute the desired elementwise divisions in the pseudo-steady-state regime $\tau_\mathrm{p}, \tau_\mathrm{z} \downarrow 0$.

Putting everything together, this gives the circuit dynamics

$$\mathbf{c}(t) = \mathbf{\Gamma}\mathbf{g}(t) \tag{B.18}$$

$$\tau_\mathrm{g}\dot{\mathbf{g}}(t) = (\mathbf{A}\mathbf{\Gamma})^\top (\mathbf{p} - \mathbf{1}) + \mathbf{\Gamma}^\top (\mathbf{z} - \boldsymbol{\lambda}) + \boldsymbol{\xi}(t) \tag{B.19}$$

$$\tau_\mathrm{p}\dot{\mathbf{p}}(t) = \mathbf{s} - \mathbf{p} \odot (\mathbf{r}_0 + \mathbf{A}\mathbf{\Gamma}\mathbf{g}) \tag{B.20}$$

$$\tau_\mathrm{z}\dot{\mathbf{z}}(t) = \boldsymbol{\alpha} - \mathbf{1} - \mathbf{z} \odot \mathbf{c}, \tag{B.21}$$

where we recall that the covariance of the zero-mean Gaussian noise process $\boldsymbol{\xi}(t)$ is

$$\mathbb{E}[\xi_j(t)\xi_{j'}(t')] = 2\tau_\mathrm{g}\delta_{jj'}\delta(t - t'). \tag{B.22}$$

We note that the two constant terms in the dynamics of $\mathbf{g}$ can be grouped into an overall leak term $-\mathbf{\Gamma}^\top(\mathbf{A}^\top \mathbf{1} + \boldsymbol{\lambda})$. As detailed in the main text, we interpret $\mathbf{p}$ as M/T cells, $\mathbf{g}$ as granule cells, and $\mathbf{z}$ as cortical feedback.

## C   Stability of the MAP fixed point

In this Appendix, we aim to get some understanding for how the introduction of the cell type $\mathbf{p}$ to represent the elementwise division affects the inference, particularly when $\tau_\mathrm{p}$ is comparable to $\tau_\mathrm{g}$. For simplicity, we specialize to the case of an exponential prior (i.e., we set $\boldsymbol{\alpha} = \mathbf{1}$), in which case the general circuit (3) simplifies to

$$\begin{aligned} \mathbf{c}(t) &= \mathbf{\Gamma}\mathbf{g}(t) \\ \tau_\mathrm{g}\dot{\mathbf{g}}(t) &= (\mathbf{A}\mathbf{\Gamma})^\top (\mathbf{p} - \mathbf{1}) - \mathbf{\Gamma}^\top \boldsymbol{\lambda} \\ \tau_\mathrm{p}\dot{\mathbf{p}}(t) &= \mathbf{s} - \mathbf{p} \odot (\mathbf{r}_0 + \mathbf{A}\mathbf{\Gamma}\mathbf{g}). \end{aligned} \tag{C.1}$$

### C.1   Analysis of a two-cell circuit

To build intuition, we first consider a circuit with $n_\mathrm{odor} = n_\mathrm{OSN} = n_\mathrm{g} = 1$:

$$\begin{aligned} c(t) &= \gamma g(t) \\ \tau_\mathrm{g}\frac{dg}{dt} &= a\gamma(p - 1) - \gamma\lambda \\ \tau_\mathrm{p}\frac{dp}{dt} &= s - p(r_0 + a\gamma g). \end{aligned} \tag{C.2}$$

In this case, it is useful to non-dimensionalize the system. Re-scale time as

$$\tilde{t} = \frac{a\gamma}{\tau_\mathrm{g}}t, \tag{C.3}$$

and define the dimensionless parameters and input

$$\tilde{\tau} = \frac{\tau_\mathrm{p}}{\tau_\mathrm{g}}, \tag{C.4}$$

$$\tilde{r}_0 = \frac{r_0}{a\gamma}, \tag{C.5}$$

$$\tilde{\lambda} = \frac{\lambda}{a}, \tag{C.6}$$

$$\tilde{s} = \frac{s}{a\gamma}. \tag{C.7}$$

Then, the dynamics are

$$\frac{dg}{d\tilde{t}} = p - 1 - \tilde{\lambda} \tag{C.8}$$

$$\tilde{\tau}\frac{dp}{d\tilde{t}} = \tilde{s} - p(\tilde{r}_0 + g). \tag{C.9}$$

These dynamics have a single fixed point at

$$g_* = \frac{\tilde{s}}{1 + \tilde{\lambda}} - \tilde{r}_0 \tag{C.10}$$

$$p_* = 1 + \tilde{\lambda}. \tag{C.11}$$

Linearizing about this fixed point, we have

$$\frac{d}{d\tilde{t}} \begin{pmatrix} \delta g \\ \delta g \end{pmatrix} = \mathbf{M} \begin{pmatrix} \delta g \\ \delta g \end{pmatrix} \tag{C.12}$$

for

$$\mathbf{M} = \frac{1}{\tilde{\tau}} \begin{pmatrix} 0 & \tilde{\tau} \\ -(1 + \tilde{\lambda}) & -(1 + \tilde{\lambda})^{-1}\tilde{s} \end{pmatrix}. \tag{C.13}$$

The eigenvalues of $\mathbf{M}$ are easily computed as

$$\Lambda_{\pm}(\mathbf{M}) = \frac{-\tilde{s} \pm \sqrt{\tilde{s}^2 - 4\tilde{\tau}(1 + \tilde{\lambda})^3}}{2\tilde{\tau}(1 + \tilde{\lambda})}. \tag{C.14}$$

For any positive $\tilde{\tau}$ and non-negative $\tilde{\lambda}$, we have

$$\mathrm{Re}\,\Lambda_{\pm}(\mathbf{M}) = \frac{-\tilde{s} \pm \mathrm{Re}\,\sqrt{\tilde{s}^2 - 4\tilde{\tau}(1 + \tilde{\lambda})^3}}{2\tilde{\tau}(1 + \tilde{\lambda})} \tag{C.15}$$

$$= \begin{cases} \dfrac{-\tilde{s} \pm \sqrt{\tilde{s}^2 - 4\tilde{\tau}(1 + \tilde{\lambda})^3}}{2\tilde{\tau}(1 + \tilde{\lambda})} & \tilde{s}^2 - 4\tilde{\tau}(1 + \tilde{\lambda})^3 > 0 \\ \dfrac{-\tilde{s}}{2\tilde{\tau}(1 + \tilde{\lambda})} & \text{otherwise,} \end{cases} \tag{C.16}$$

hence it is easy to see that the real parts of both of these eigenvalues are strictly negative so long as $\tilde{s} > 0$, meaning that the system is stable. If $\tilde{s} = 0$—which should be exceedingly rare if the OSNs have a baseline rate—then

$$\Lambda_{\pm}(\mathbf{M})\Big|_{\tilde{s}=0} = \pm 2i\sqrt{\frac{1 + \tilde{\lambda}}{\tilde{\tau}}}, \tag{C.17}$$

and there can be oscillations. Therefore, a linear stability analysis suggests that the MAP fixed point of this two-cell circuit should be stable even for large $\tilde{\tau}$ in the presence of a non-zero input, though we expect the relaxation timescales to grow as $\tilde{\tau}$ becomes larger.

## C.2 Analysis of the full circuit

We now consider the full MAP circuit

$$\begin{aligned} \mathbf{c}(t) &= \mathbf{\Gamma}\mathbf{g}(t) \\ \tau_{\mathrm{g}}\dot{\mathbf{g}}(t) &= (\mathbf{A}\mathbf{\Gamma})^{\top}(\mathbf{p} - \mathbf{1}) - \mathbf{\Gamma}^{\top}\boldsymbol{\lambda} \\ \tau_{\mathrm{p}}\dot{\mathbf{p}}(t) &= \mathbf{s} - \mathbf{p} \odot (\mathbf{r}_0 + \mathbf{A}\mathbf{\Gamma}\mathbf{g}). \end{aligned} \tag{C.18}$$

We assume that $n_{\mathrm{OSN}} < n_{\mathrm{odor}} \leq n_{\mathrm{g}}$ and that $\mathbf{A}\mathbf{\Gamma}$ is of full row rank, i.e., it has rank $n_{\mathrm{OSN}}$. We recall that $\mathbf{g} \in \mathbb{R}_+^{n_{\mathrm{g}}}$ and $\mathbf{p} \in \mathbb{R}_+^{n_{\mathrm{g}}}$. Therefore, the dynamics of $\mathbf{g}$ span only an $n_{\mathrm{odor}}$-dimensional subspace, and, in particular, the $\mathbf{p}$-dependent term affects only an $n_{\mathrm{OSN}}$-dimensional subspace.

We start by observing that the dynamics of $\mathbf{p}$ depend on $\mathbf{g}$ only through

$$\mathbf{q} \equiv \mathbf{A}\mathbf{\Gamma}\mathbf{g}. \tag{C.19}$$

Importantly, if $\mathbf{A}\mathbf{\Gamma}$ is positivity-preserving, then $\mathbf{q} \in \mathbb{R}_+^{n_{\mathrm{OSN}}}$. Then, we have the closed dynamics

$$\begin{aligned} \tau_{\mathrm{g}}\dot{\mathbf{q}} &= (\mathbf{A}\mathbf{\Gamma}\mathbf{\Gamma}^{\top}\mathbf{A}^{\top})(\mathbf{p} - \mathbf{1}) - \mathbf{A}\mathbf{\Gamma}\mathbf{\Gamma}^{\top}\boldsymbol{\lambda} \\ \tau_{\mathrm{p}}\dot{\mathbf{p}} &= \mathbf{s} - \mathbf{p} \odot (\mathbf{r}_0 + \mathbf{q}) \end{aligned} \tag{C.20}$$

for $(\mathbf{q}, \mathbf{p}) \in \mathbb{R}_+^{2n_{\mathrm{OSN}}}$. The fixed point of this system is of course determined by the conditions $d(\mathbf{q}, \mathbf{p})/dt = \mathbf{0}$, which gives

$$(\mathbf{A}\mathbf{\Gamma}\mathbf{\Gamma}^\top \mathbf{A}^\top)(\mathbf{p}^* - \mathbf{1}) = \mathbf{A}\mathbf{\Gamma}\mathbf{\Gamma}^\top \boldsymbol{\lambda} \tag{C.21}$$

$$\mathbf{p}^* \odot (\mathbf{r}_0 + \mathbf{q}^*) = \mathbf{s} \tag{C.22}$$

subject to the non-negativity constraints. By our assumptions on the rank of $\mathbf{A}\mathbf{\Gamma}$, the symmetric matrix $\mathbf{A}\mathbf{\Gamma}\mathbf{\Gamma}^\top \mathbf{A}^\top$ is positive-definite and thus can be inverted to solve the first condition for $\mathbf{p}^*$:

$$\mathbf{p}^* = \mathbf{1} + (\mathbf{A}\mathbf{\Gamma}\mathbf{\Gamma}^\top \mathbf{A}^\top)^{-1}\mathbf{A}\mathbf{\Gamma}\mathbf{\Gamma}^\top \boldsymbol{\lambda}. \tag{C.23}$$

For the second condition to be satisfied, we can see that the elements of $\mathbf{p}^*$ must be strictly positive at the fixed point, which gives a self-consistency condition on $\mathbf{A}$, $\mathbf{\Gamma}$, and $\boldsymbol{\lambda}$. Assuming that this holds, we then have

$$\mathbf{q}^* = \mathbf{s} \oslash \mathbf{p}^* - \mathbf{r}_0, \tag{C.24}$$

which again gives a self-consistency condition as non-negativity is required.

Assuming these conditions hold, we can then linearize the dynamics about the fixed point. For convenience, we non-dimensionalize time through

$$\tilde{t} = \frac{t}{\tau_{\mathrm{g}}}, \tag{C.25}$$

and

$$\tilde{\tau} = \frac{\tau_{\mathrm{p}}}{\tau_{\mathrm{g}}}, \tag{C.26}$$

which yields the linearized dynamics

$$\frac{d}{d\tilde{t}}\begin{pmatrix} \delta\mathbf{q} \\ \delta\mathbf{p} \end{pmatrix} = \mathbf{M}\begin{pmatrix} \delta\mathbf{q} \\ \delta\mathbf{p} \end{pmatrix} \tag{C.27}$$

for

$$\mathbf{M} = \begin{pmatrix} \mathbf{0} & (\mathbf{A}\mathbf{\Gamma}\mathbf{\Gamma}^\top \mathbf{A}^\top) \\ -\tilde{\tau}^{-1}\operatorname{diag}(\mathbf{p}^*) & -\tilde{\tau}^{-1}\operatorname{diag}(\mathbf{r}_0 + \mathbf{q}^*) \end{pmatrix} \tag{C.28}$$

$$= \begin{pmatrix} \mathbf{0} & (\mathbf{A}\mathbf{\Gamma}\mathbf{\Gamma}^\top \mathbf{A}^\top) \\ -\tilde{\tau}^{-1}\operatorname{diag}(\mathbf{p}^*) & -\tilde{\tau}^{-1}\operatorname{diag}(\mathbf{s} \oslash \mathbf{p}^*) \end{pmatrix}. \tag{C.29}$$

Using the fact that the diagonal matrices $\operatorname{diag}(\mathbf{p}^*)$ and $\operatorname{diag}(\mathbf{s} \oslash \mathbf{p}^*)$ commute, the characteristic polynomial of $\mathbf{M}$ is

$$\det(\mathbf{M} - \mu\mathbf{I}_{2n_{\mathrm{OSN}}}) = \det\begin{pmatrix} \mu\mathbf{I}_{n_{\mathrm{OSN}}} & -(\mathbf{A}\mathbf{\Gamma}\mathbf{\Gamma}^\top \mathbf{A}^\top) \\ \tilde{\tau}^{-1}\operatorname{diag}(\mathbf{p}^*) & \tilde{\tau}^{-1}\operatorname{diag}(\mathbf{s} \oslash \mathbf{p}^*) + \mu\mathbf{I}_{n_{\mathrm{OSN}}} \end{pmatrix} \tag{C.30}$$

$$= \det[\mu^2\mathbf{I}_{n_{\mathrm{OSN}}} + \mu\tilde{\tau}^{-1}\operatorname{diag}(\mathbf{s} \oslash \mathbf{p}^*) + \tilde{\tau}^{-1}(\mathbf{A}\mathbf{\Gamma}\mathbf{\Gamma}^\top \mathbf{A}^\top)\operatorname{diag}(\mathbf{p}^*)]. \tag{C.31}$$

One case that is particularly easy to solve is when the symmetric positive-definite matrix $\mathbf{A}\mathbf{\Gamma}\mathbf{\Gamma}^\top \mathbf{A}^\top$ is in fact diagonal, with positive diagonal entries $a_j$. Then, we have

$$\det(\mathbf{M} - \mu\mathbf{I}_{2n_{\mathrm{OSN}}}) = \prod_{j=1}^{n_{\mathrm{OSN}}}\left[\mu^2 + \frac{s_j}{\tilde{\tau}p_j^*}\mu + \frac{a_j p_j^*}{\tilde{\tau}}\right], \tag{C.32}$$

meaning that the eigenvalues of $\mathbf{M}$ are

$$\mu_{j,\pm} = \frac{1}{2}\left[-\frac{s_j}{\tilde{\tau}p_j^*} \pm \sqrt{\left(\frac{s_j}{\tilde{\tau}p_j^*}\right)^2 - 4\frac{a_j p_j^*}{\tilde{\tau}}}\right], \tag{C.33}$$

which under the given assumptions always have strictly positive real part for non-negative inputs.

Now, more generally, assume that $\mathbf{M}$ is diagonalizable, and let $\mathbf{m}$ be a un-normalized eigenvector of $\mathbf{M}$ with eigenvalue $\mu$. Then, writing

$$\mathbf{m} = \begin{pmatrix} \mathbf{u} \\ \mathbf{v} \end{pmatrix} \tag{C.34}$$

for $\mathbf{u}, \mathbf{v} \in \mathbb{C}^{n_{\text{OSN}}}$, the eigenvector condition

$$\mathbf{M}\mathbf{m} = \mu\mathbf{m} \tag{C.35}$$

implies that $\mathbf{u}$ and $\mathbf{v}$ satisfy

$$\frac{1}{\tau_{\text{g}}}(\mathbf{A}\mathbf{\Gamma}\mathbf{\Gamma}^\top\mathbf{A}^\top)\mathbf{v} = \mu\mathbf{u} \tag{C.36}$$

$$-\frac{1}{\tau_{\text{p}}}\operatorname{diag}(\mathbf{p}^*)\mathbf{u} - \frac{1}{\tau_{\text{p}}}\operatorname{diag}(\mathbf{s} \oslash \mathbf{p}^*)\mathbf{v} = \mu\mathbf{v}. \tag{C.37}$$

Assuming that $\mu$ is non-zero, we can solve the first equation for $\mathbf{u}$ and then substitute the result into the second to obtain a quadratic eigenproblem for $\mu$ and $\mathbf{v}$:

$$\mu^2\mathbf{v} + \mu\frac{1}{\tau_{\text{p}}}\operatorname{diag}(\mathbf{s} \oslash \mathbf{p}^*)\mathbf{v} + \frac{1}{\tau_{\text{p}}\tau_{\text{g}}}\operatorname{diag}(\mathbf{p}^*)(\mathbf{A}\mathbf{\Gamma}\mathbf{\Gamma}^\top\mathbf{A}^\top)\mathbf{v} = 0. \tag{C.38}$$

We will not attempt to solve this eigenproblem, but will instead attempt to extract information about possible values of $\mu$ for a fixed $\mathbf{v}$. Suppose (without loss of generality given the assumption that it is nonzero, as otherwise we may divide by its norm) that $\mathbf{v}$ is a unit vector. Then, acting with $\mathbf{v}^\dagger$ from the left, we have

$$\mu^2 + \frac{a}{\tau_{\text{p}}}\mu + \frac{b}{\tau_{\text{p}}\tau_{\text{g}}} = 0. \tag{C.39}$$

where we define the coefficients

$$a \equiv \mathbf{v}^\dagger\operatorname{diag}(\mathbf{s} \oslash \mathbf{p}^*)\mathbf{v} \tag{C.40}$$

and

$$b \equiv \mathbf{v}^\dagger\operatorname{diag}(\mathbf{p}^*)(\mathbf{A}\mathbf{\Gamma}\mathbf{\Gamma}^\top\mathbf{A}^\top)\mathbf{v}. \tag{C.41}$$

So long as $\mathbf{s}$ and $\mathbf{p}^*$ are positive, $a$ is real and positive. Let

$$\tilde{\boldsymbol{\lambda}} \equiv (\mathbf{A}\mathbf{\Gamma}\mathbf{\Gamma}^\top\mathbf{A}^\top)^{-1}\mathbf{A}\mathbf{\Gamma}\mathbf{\Gamma}^\top\boldsymbol{\lambda} \tag{C.42}$$

such that

$$\mathbf{p}^* = \mathbf{1} + \tilde{\boldsymbol{\lambda}}. \tag{C.43}$$

By assumption, the elements of $\tilde{\boldsymbol{\lambda}}$ are strictly greater than $-1$. Then,

$$\operatorname{Re}(b) = \mathbf{v}^\dagger\frac{\operatorname{diag}(\mathbf{p}^*)(\mathbf{A}\mathbf{\Gamma}\mathbf{\Gamma}^\top\mathbf{A}^\top) + (\mathbf{A}\mathbf{\Gamma}\mathbf{\Gamma}^\top\mathbf{A}^\top)\operatorname{diag}(\mathbf{p}^*)}{2}\mathbf{v} \tag{C.44}$$

$$= \mathbf{v}^\dagger(\mathbf{A}\mathbf{\Gamma}\mathbf{\Gamma}^\top\mathbf{A}^\top)\mathbf{v} + \mathbf{v}^\dagger\frac{\operatorname{diag}(\tilde{\boldsymbol{\lambda}})(\mathbf{A}\mathbf{\Gamma}\mathbf{\Gamma}^\top\mathbf{A}^\top) + (\mathbf{A}\mathbf{\Gamma}\mathbf{\Gamma}^\top\mathbf{A}^\top)\operatorname{diag}(\tilde{\boldsymbol{\lambda}})}{2}\mathbf{v}, \tag{C.45}$$

while

$$\operatorname{Im}(b) = \mathbf{v}^\dagger\frac{\operatorname{diag}(\mathbf{p}^*)(\mathbf{A}\mathbf{\Gamma}\mathbf{\Gamma}^\top\mathbf{A}^\top) - (\mathbf{A}\mathbf{\Gamma}\mathbf{\Gamma}^\top\mathbf{A}^\top)\operatorname{diag}(\mathbf{p}^*)}{2i}\mathbf{v} \tag{C.46}$$

$$= \mathbf{v}^\dagger\frac{\operatorname{diag}(\tilde{\boldsymbol{\lambda}})(\mathbf{A}\mathbf{\Gamma}\mathbf{\Gamma}^\top\mathbf{A}^\top) - (\mathbf{A}\mathbf{\Gamma}\mathbf{\Gamma}^\top\mathbf{A}^\top)\operatorname{diag}(\tilde{\boldsymbol{\lambda}})}{2i}\mathbf{v}. \tag{C.47}$$

Then, solving the quadratic equation for $\mu$, we have

$$\mu = \frac{1}{\tau_{\text{p}}}\frac{-a \pm \sqrt{a^2 - 4\tilde{\tau}b}}{2}, \tag{C.48}$$

for

$$\tilde{\tau} \equiv \frac{\tau_{\mathrm{p}}}{\tau_{\mathrm{g}}}. \tag{C.49}$$

This gives

$$\mathrm{Re}(\mu) = -\frac{a}{2\tau_{\mathrm{p}}} \pm \frac{1}{2\tau_{\mathrm{p}}} \, \mathrm{Re} \, \sqrt{a^2 - 4\tilde{\tau} b} \tag{C.50}$$

$$= -\frac{a}{2\tau_{\mathrm{p}}} \pm \frac{1}{2\tau_{\mathrm{p}}} \sqrt{\frac{\sqrt{(a^2 - 4\tilde{\tau}\,\mathrm{Re}(b))^2 + 16\tilde{\tau}^2\,\mathrm{Im}(b)^2} + a^2 - 4\tilde{\tau}\,\mathrm{Re}(b)}{2}}. \tag{C.51}$$

If $\mathrm{Im}(b) = 0$, then we are in the same situation as we were for the two-neuron circuit, and the fixed point is thus stable.

More generally, for the system to be stable we want to have

$$a > \sqrt{\frac{\sqrt{(a^2 - 4\tilde{\tau}\,\mathrm{Re}(b))^2 + 16\tilde{\tau}^2\,\mathrm{Im}(b)^2} + a^2 - 4\tilde{\tau}\,\mathrm{Re}(b)}{2}}. \tag{C.52}$$

For either sign of $a^2 - 4\tilde{\tau}\,\mathrm{Re}(b)$, one can show that this holds provided that

$$\mathrm{Re}(b) > 0 \tag{C.53}$$

and

$$\sqrt{\tilde{\tau}}|\,\mathrm{Im}(b)| < a\sqrt{\mathrm{Re}(b)}. \tag{C.54}$$

This amounts to a combined condition on the effective prior strength $\tilde{\boldsymbol{\lambda}}$ and the relative time constants. Therefore, for sufficiently small $\tilde{\boldsymbol{\lambda}}$, we have stability for a broad range of $\tau_{\mathrm{p}}$.

How can we go from stability of the $(\mathbf{q}, \mathbf{p})$ circuit to stability of the full circuit? Heuristically, this follows by decomposing the dynamics of $\mathbf{g}$ in terms of the different subspaces. We will not attempt to do this rigorously, as our goal is only to get a rough sense of how the system should behave. By the fact that their dimensions are strictly ordered, we know that $\mathrm{span}(\boldsymbol{\Gamma}^\top) \supset \mathrm{span}[(\mathbf{A}\boldsymbol{\Gamma})^\top]$. We can first exclude components outside $\mathrm{span}(\boldsymbol{\Gamma}^\top)$, as they will be unchanged by the dynamics and do not affect the concentration estimate. Recalling the non-negativity constraint, components in $\mathrm{span}(\boldsymbol{\Gamma}^\top) \setminus \mathrm{span}[(\mathbf{A}\boldsymbol{\Gamma})^\top]$ will decay linearly to zero. Then, the components in $\mathrm{span}[(\mathbf{A}\boldsymbol{\Gamma})^\top]$ should be controllable using the argument for the $(\mathbf{q}, \mathbf{p})$ space. Giving a fully rigorous analysis of the conditions under which the MAP fixed point is stable will be an interesting avenue for future investigation.

## D   Derivation for the Gaussian circuit model

To compare the result of the state dependent inhibition due to estimating the MAP for a Poisson noise model with the fixed inhibition resulting from Gaussian noise we build an alternative circuit model. This circuit still shares the separation into two cell types, but the granule cells $\mathbf{g}$ converge on the solution for a Gaussian noise model.

Our starting point is an isotropic Gaussian likelihood

$$\mathbf{s} \,|\, \mathbf{c} \sim \mathcal{N}(\mathbf{r}_0 + \mathbf{A}\mathbf{c}, \sigma^2 \mathbf{I}_{n_{\mathrm{OSN}}}) \tag{D.1}$$

variance $\sigma^2$, and an exponential prior

$$\mathbf{c} \sim \mathrm{Exp}(\boldsymbol{\lambda}). \tag{D.2}$$

Gradient ascent on the resulting log-posterior over $\mathbf{c}$ leads to the dynamics:

$$\dot{\mathbf{c}}(t) = \frac{1}{\sigma^2}\mathbf{A}^\top[\mathbf{s} - (\mathbf{r}_0 + \mathbf{A}\mathbf{c})] - \boldsymbol{\lambda}, \tag{D.3}$$

Distributing the code such that $\mathbf{c}(t) = \boldsymbol{\Gamma}\mathbf{g}$ and splitting the inference as we did in the Poisson case gives the circuit dynamics

$$\mathbf{c}(t) = \boldsymbol{\Gamma}\mathbf{g}(t)$$

$$\tau_{\mathrm{g}}\dot{\mathbf{g}}(t) = \frac{1}{\sigma^2}(\mathbf{A}\boldsymbol{\Gamma})^\top\mathbf{p} - \boldsymbol{\Gamma}^\top\boldsymbol{\lambda} \tag{D.4}$$

$$\tau_{\mathrm{p}}\dot{\mathbf{p}}(t) = -\mathbf{p} + \mathbf{s} - (\mathbf{r}_0 + \mathbf{A}\boldsymbol{\Gamma}\mathbf{g}).$$

Unlike in the Poisson case, the inhibition onto the projection cells (mitral cells) is not gated by their activity, leading to the fixed offset shown in Fig. 2B.

# E  Extending the sampling circuit to incorporate an $L_0$ prior

In this Appendix, we consider the possibility of extending our circuit model to incorporate an $L_0$ spike-and-slab prior

$$p(c_i) = \varpi e^{-\lambda c_i} + (1 - \varpi)\delta(c_i), \tag{E.1}$$

where $\varpi$ is the probability that the odor is present and $\lambda$ is the rate of the exponential prior on concentrations given that the odor is present. To sample from the resulting posterior using Langevin dynamics, we will follow the approach of Fang et al. [115], who developed a Langevin algorithm to perform sparse coding with this prior given a Gaussian likelihood. This approach only works in the sampling setting; it cannot be applied to MAP estimation because the Dirac mass at $c_i = 0$ means that the MAP estimate will always vanish.

In this approach, we define an auxiliary variable $\mathbf{u}$ that is mapped to concentration estimates $\mathbf{c}$ via element-wise soft thresholding:

$$\mathbf{c} = f(\mathbf{u}), \tag{E.2}$$

where

$$f(u) = \begin{cases} 0 & u < u_0 \\ u - u_0 & u \geq u_0 \end{cases} \tag{E.3}$$

is the soft-thresholding function for threshold

$$u_0 = -\frac{1}{\lambda}\log\varpi. \tag{E.4}$$

We then posit the following Langevin dynamics for $\mathbf{u}$, given an observation $\mathbf{s}$:

$$\dot{\mathbf{u}}(t) = \left[\boldsymbol{\nabla}_{\mathbf{c}}\log p(\mathbf{s}\,|\,\mathbf{c})\right]_{\mathbf{c}=f(|\mathbf{u}|)} \odot\,\Theta(|\mathbf{u}| - \mathbf{u}_0) - \lambda\,\mathrm{sign}(\mathbf{u}) + \boldsymbol{\eta}(t) \tag{E.5}$$

with no constraint on the sign of $\mathbf{u}$, where $\boldsymbol{\eta}(t)$ is a white Gaussian noise process. Here, all nonlinearities are applied element-wise, and $\mathbf{u}_0 = u_0\mathbf{1}$ is a vector with all elements equal to the threshold $u_0$. Fang et al. [115] argue that the stationary distribution induced on $\mathbf{c} = f(\mathbf{u})$ by these dynamics should be the desired posterior with $L_0$ prior.

Using the Poisson likelihood gradient as computed before, we have

$$\dot{\mathbf{u}}(t) = \{\mathbf{A}^\top[\mathbf{s} \oslash (\mathbf{r}_0 + \mathbf{A}f(|\mathbf{u}|)) - \mathbf{1}]\} \odot\,\Theta(|\mathbf{u}| - \mathbf{u}_0) - \lambda\,\mathrm{sign}(\mathbf{u}) + \boldsymbol{\eta}(t) \tag{E.6}$$

At this stage, we can see that the soft thresholding has picked out a preferred basis. If we apply the complete recipe in a way analogous to what we did in Appendix B by writing

$$\mathbf{u} = \boldsymbol{\Gamma}\mathbf{g}, \tag{E.7}$$

we will have

$$\tau_{\mathrm{g}}\dot{\mathbf{g}}(t) = \boldsymbol{\Gamma}^\top\left[\{\mathbf{A}^\top[\mathbf{s} \oslash (\mathbf{r}_0 + \mathbf{A}f(|\boldsymbol{\Gamma}\mathbf{g}|)) - \mathbf{1}]\} \odot\,\Theta(|\boldsymbol{\Gamma}\mathbf{g}| - \mathbf{u}_0) - \lambda\,\mathrm{sign}(\boldsymbol{\Gamma}\mathbf{g})\right] + \boldsymbol{\xi}(t), \tag{E.8}$$

where $\boldsymbol{\xi}(t)$ is a zero-mean Gaussian noise process with covariance

$$\mathbb{E}[\xi_j(t)\xi_{j'}(t')] = 2\tau_{\mathrm{g}}\delta_{jj'}\delta(t - t'). \tag{E.9}$$

Here, we cannot simply regroup terms; if we introduce an additional cell type $\mathbf{p}$ as before to compute the division we will have a sort of effective weight matrix

$$\sum_k \Gamma_{ki}A_{jk}\Theta(|u_k| - u_0) \tag{E.10}$$

for the $\mathbf{p}$-to-$\mathbf{g}$ connections, and an entirely different coupling $\mathbf{A}f(|\boldsymbol{\Gamma}\mathbf{g}|)$ for the $\mathbf{g}$-to-$\mathbf{p}$ connections. Concretely, introducing $\mathbf{p}$ as before, we have the system

$$\mathbf{c}(t) = f(|\boldsymbol{\Gamma}\mathbf{g}(t)|) \tag{E.11}$$

$$\tau_{\mathrm{g}}\dot{\mathbf{g}}(t) = \boldsymbol{\Gamma}^\top\left[[\mathbf{A}^\top(\mathbf{p} - \mathbf{1})] \odot\,\Theta(|\boldsymbol{\Gamma}\mathbf{g}| - \mathbf{u}_0) - \lambda\,\mathrm{sign}(\boldsymbol{\Gamma}\mathbf{g})\right]dt + \boldsymbol{\xi}(t) \tag{E.12}$$

$$\tau_{\mathrm{p}}\dot{\mathbf{p}}(t) = \mathbf{s} - \mathbf{p} \odot [\mathbf{r}_0 + \mathbf{A}f(|\boldsymbol{\Gamma}\mathbf{g}|)]. \tag{E.13}$$

This issue illustrates the limitations of the simple, linear approach to distributing the neural code used in Appendix B. In particular, the nonlinearity in some sense picks out a preferred basis, meaning that we can no longer perform a simple linear change of coordinates to distribute the code.

# F  Experimental details and affinity matrix fitting

Here we briefly describe the experimental procedures used to collect the data used to fit the parameters of the affinity matrix $\mathbf{A}$ in Fig. 1. The full experimental methods are described in the paper that first presented the data, Zak et al. [94]. All the experiments were performed in accordance with the guidelines set by the National Institutes of Health and approved by the Institutional Animal Care and Use Committee at Harvard University.

## F.1  *In vivo* recordings from mouse OB

Adult ($> 8$ weeks) OMP-GCaMP3 mice of both sexes were used in this study. A craniotomy was performed to provide optical access to olfactory sensory neuron axon terminals in both olfactory bulbs. A custom-built two-photon microscope was used for in vivo imaging. Images were acquired at 16-bit resolution and 4-8 frames/s. The pixel size was $0.6\ \mu m$ and the fields of view were $720 \times 720\ \mu m$. Monomolecular odorants (Allyl butyrate, Ethyl valerate, Methyl tiglate, and Isobutyl propionate) were used as stimuli and delivered by a 16-channel olfactometer controlled by custom-written software in LabView. For the odorant concentration series, the initial odorant concentration was between $0.08\% - 80\%(v/v)$ in mineral oil and further diluted 16 times with air. The relative odorant concentrations were measured by a photoionization detector, then normalized to the largest detected signal for each odorant. For all experiments, the airflow to the animal was held constant at $100mL/min$, and odorants were injected into a carrier stream. Each odorant concentration was delivered 2–6 times in pseudorandom order.

Images were processed using both custom and available MATLAB scripts. Motion artifact compensation and denoising were done using *NoRMcorre* [138]. The $\Delta F/F$ signal was calculated by finding the peak signal following odorant onset and averaging with the two adjacent points. To account for changes in respiration and anesthesia depth, correlated variability was corrected for [139]. Thresholds for classifying responding ROIs were determined from a noise distribution of blank (no odorant) trials from which three standard deviations were used for responses.

## F.2  Gamma distribution fitting

In order to fit the affinity matrix $\mathbf{A}$ to the experimentally recorded data, we normalized the response of each glomerulus to its maximum response across the panel of 32 odorants. We then vectorized the resulting matrix and fitted a $\mathrm{Gamma}$ distribution using the `gamfit` function in MATLAB. As mentioned in the main text, this resulted in a $\mathrm{Gamma}(0.37, 0.36)$ distribution.

# G  Numerical methods and supplemental figures

All numerical simulations were performed using MATLAB 9.13 (R2022b, The MathWorks, Natick, MA, USA) either on desktop workstations (CPU: Intel i9-9900K or Xeon W-2145, 64GB RAM) or on the Harvard University FASRC Cannon HPC cluster (https://www.rc.fas.harvard.edu). Our simulations were not computationally intensive, and required around that 6000 CPU-hours of total compute time.

## G.1  State-dependent inhibition simulations

For the simulations to highlight the signatures of state-dependent inhibition of the Poisson network and compare its behavior with the experimental observation by Arevian et al. [95] we use a reduced network with $n_{OSN} = 2$ and $n_g = 10$. We chose this reduced circuit both to reduce computational costs and to match the experimental parameters of the *in-vitro* experiment. The other parameters of the simulation where as follows: $\tau_p = 0.02$, $\tau_g = 0.03$, $r_0 = 10$, $\lambda = 2$, $dt = 1e^{-4}$. The simulation ran for $600\ ms$, stimulation was applied starting at $t_{start} = 100\ ms$ and ended at $t_{end} = 500\ ms$. We stimulated the principal mitral cell $MC_A$ with 80 equally spaced values in $s_A \in [1, 400]$. When the second mitral cell was active, its stimulation was set at $s_B = 80$. The entries of the $\Gamma$ matrix were sampled from a normal distribution on which we applied a mask such that only $25\%$ of its entries were non-zero. We sampled 32 'pairs' of cells by resampling the $\Gamma$ matrix for each 'pair' of cells. The simulation for the Gaussian circuit used the same parameters except the dynamics followed those

from equation D.4. For plotting purposes, we normalized the range of firing in Figure 2B to the maximum firing rate for each circuit model.

## G.2 Capacity simulations

In our capacity simulations, we use $n_{\text{OSN}} = 300$ and $n_{\text{odor}} = 500, 1000,$ or $2000$. We take the odor stimulus to be a rectangular pulse, with varying numbers of randomly-selected odors appearing at concentration $c_j = 40$. As elsewhere, we take $\tau_p = 20$ ms [91] and $\tau_g = 30$ ms [92] to match experiment. We set the baseline rate to be $r_0 = 1$ and the prior mean to be $\lambda = 1$; based on some experimentation our results appear relatively insensitive to small variations in these choices. We integrate the MAP circuit dynamics (3) using the forward Euler method with timestep $\Delta t = 10^{-4}$ s. To determine which odors the model estimates as being present at a given timepoint, we simply check which concentration estimates exceed 20 at that time.

As discussed in the main text, we consider three variants of the MAP circuit, defined by different choices of the matrix $\mathbf{\Gamma}$:

- For the one-to-one code, we let $n_{\text{g}} = n_{\text{odor}}$, and simply set $\mathbf{\Gamma} = \mathbf{I}_{n_{\text{odor}}}/N(A_{ij})$.
- For the naïvely distributed code, we let $n_{\text{g}} = 5n_{\text{odor}}$, and choose $\mathbf{\Gamma}$ as follows: We sample a random matrix $\mathbf{Q} \in \mathbb{R}^{n_{\text{odor}} \times n_{\text{g}}}$ with $\mathbf{Q}\mathbf{Q}^\top = \mathbf{I}_{n_{\text{odor}}}$ by drawing a Gaussian matrix and orthogonalizing its rows. Then, we define $\mathbf{\Gamma} = \mathbf{Q}/N\{|(\mathbf{AQ})_{ij}|\}$.
- For the geometry-aware code, we let $n_{\text{g}} = 5n_{\text{odor}}$, and choose $\mathbf{\Gamma}$ as follows: We sample a random matrix $\mathbf{Q} \in \mathbb{R}^{n_{\text{odor}} \times n_{\text{g}}}$ with $\mathbf{Q}\mathbf{Q}^\top = \mathbf{I}_{n_{\text{odor}}}$ by drawing a Gaussian matrix and orthogonalizing its rows. Then, we compute an approximate inverse square root of the low-rank matrix $\mathbf{A}^\top\mathbf{A}$ as $\mathbf{B} = (\mathbf{A}^\top\mathbf{A} + a\mathbf{I}_{n_{\text{odor}}})^{-1/2}$ for a small positive regularizing constant $a$. In our simulations, we set $a = 0.5$; we find empirically that our results are not substantially sensitive to small variations in $a$. Finally, we let $\mathbf{\Gamma} = \mathbf{BQ}/N\{|(\mathbf{ABQ})_{ij}|\}$.

where $N$ is a normalization function defined as

$$N(A) = \max(A) \cdot \frac{\sqrt{n_g}}{C}$$

with $C = 50$ identified as a reasonable choice. Our normalization convention for $\mathbf{\Gamma}$ is motivated by the idea that, in biology, the strength of individual synapses should be bounded. Moreover, changing the overall scale of the synaptic weights in our model corresponds to changing the effective time constant of the dynamics, which can produce a trivial speedup.

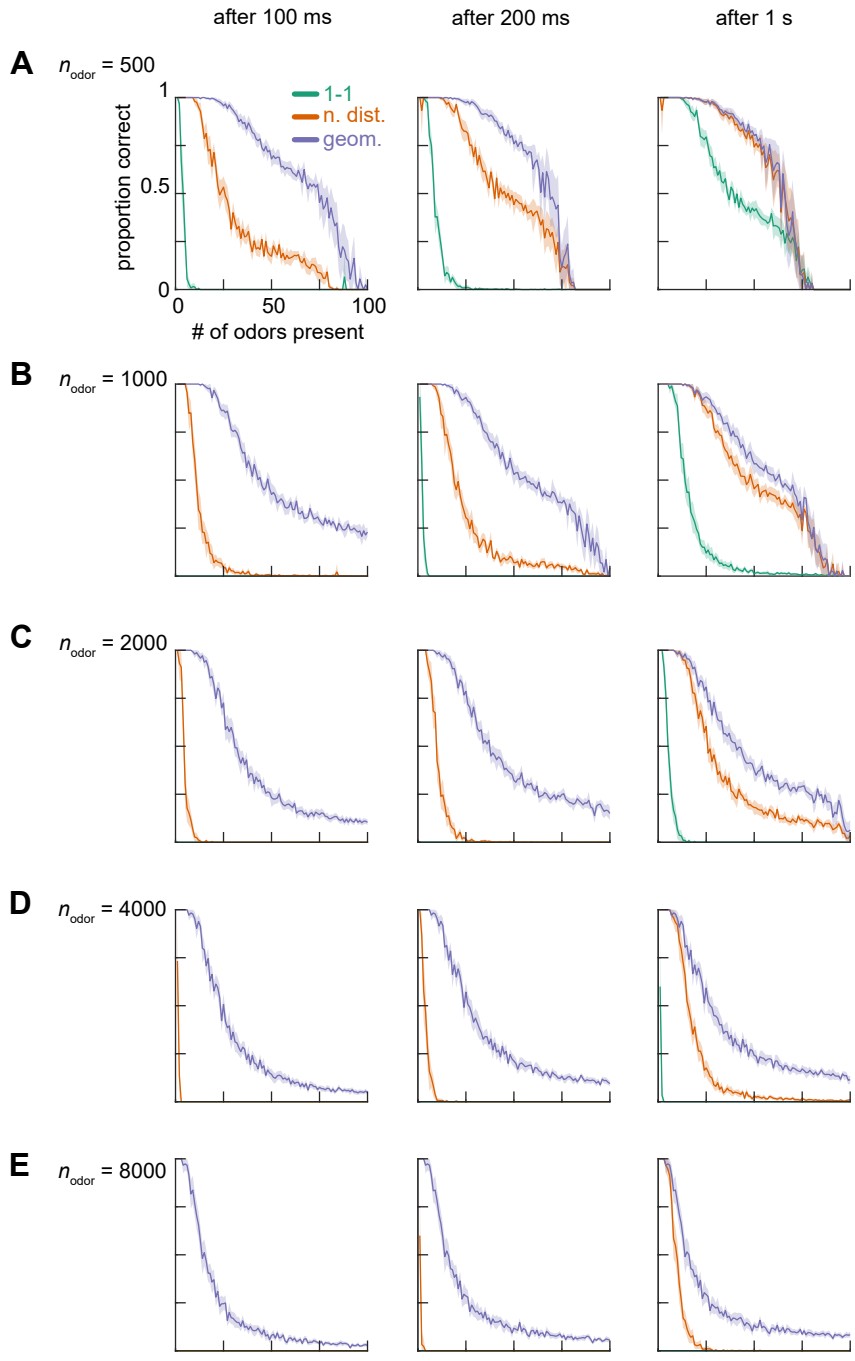

Figure G.1: **A**. Fraction of odors correctly detected within 100 ms (*left*), 200 ms (*center*), and 1 s (*right*) after odor onset as a function of the number of odors present, with $n_{\text{odor}} = 500$ possible odors, for models with one-to-one, naïvely distributed, and geometry-aware codes. **B**. As in **A**, but for $n_{\text{odor}} = 1000$. This matches Fig. 3B. **C**, **D**, **E**. As in **A**, but for $n_{\text{odor}} = 2000$, 4000, or 8000 possible odorants, respectively. Shaded patches show $\pm 1.96$ SEM over realizations throughout.

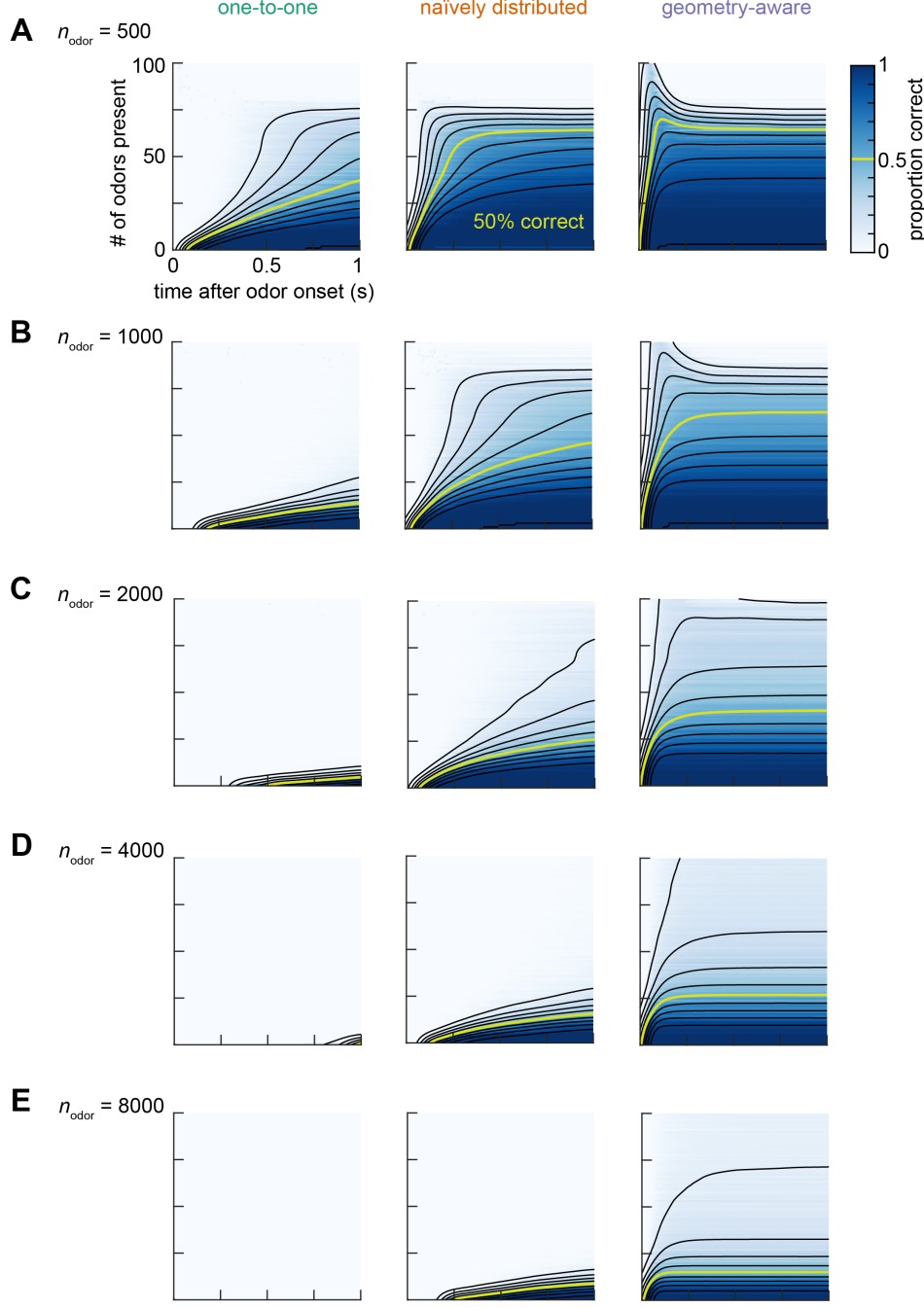

Figure G.2: **A**. Heatmap with overlaid smoothed contours of the fraction of odors correctly detected as a function of number of present odors and time window out of a panel of $n_{\text{odor}} = 500$ possible odors for models with one-to-one (*left*), naïvely distributed (*center*), and geometry-aware (*right*) codes. **B**. As in **A**, but for $n_{\text{odor}} = 1000$. This matches Fig. 3B. **C**, **D**, **E**. As in **A**, but for $n_{\text{odor}} = 2000, 4000$, or 8000 possible odorants, respectively.

### G.3 Sampling simulations

In our sampling simulations, we use $n_{\text{OSN}} = 300$, $n_{\text{odor}} = 500$ or $1000$, and $n_{\text{g}} = 5n_{\text{odor}}$. Here, the odor stimulus was composed of two rectangular pulses: At time $t_{\text{low}} = 0$ s, five "low" odors appear at concentration 10, and remain present at that concentration until time $t = 2$ s. Then, at time $t_{\text{high}} = 1$ s, five "high" odors appear at concentration 40, and remain present until time $t_{\text{off}} = 2$ s. As in our capacity simulations, we take $\tau_p = 20$ ms [91] and $\tau_g = 30$ ms [92] to match experiment, and we set the baseline rate to be $r_0 = 1$ and the prior mean to be $\lambda = 1$. As for our capacity results, our results appear relatively insensitive to small variations in these choices. We integrate the sampling circuit dynamics (4) using the forward Euler-Maruyama method with timestep $\Delta t = 10^{-5}$ s.

As in our capacity simulations, in Fig. 4 we choose $\boldsymbol{\Gamma}$ as follows:

- For the one-to-one code, we let $n_{\text{g}} = n_{\text{odor}}$, and simply set $\boldsymbol{\Gamma} = \mathbf{I}_{n_{\text{odor}}} / \max(A_{ij})$.
- For the naïvely distributed code, we let $n_{\text{g}} = 5n_{\text{odor}}$, and choose $\boldsymbol{\Gamma}$ as follows: We sample a random matrix $\mathbf{Q} \in \mathbb{R}^{n_{\text{odor}} \times n_{\text{g}}}$ with $\mathbf{Q}\mathbf{Q}^\top = \mathbf{I}_{n_{\text{odor}}}$ by drawing a Gaussian matrix and orthogonalizing its rows. Then, we define $\boldsymbol{\Gamma} = \mathbf{Q} / \max\{|(\mathbf{A}\mathbf{Q})_{ij}|\}$.
- For the geometry-aware code, we let $n_{\text{g}} = 5n_{\text{odor}}$, and choose $\boldsymbol{\Gamma}$ as follows: We sample a random matrix $\mathbf{Q} \in \mathbb{R}^{n_{\text{odor}} \times n_{\text{g}}}$ with $\mathbf{Q}\mathbf{Q}^\top = \mathbf{I}_{n_{\text{odor}}}$ by drawing a Gaussian matrix and orthogonalizing its rows. Then, we compute an approximate inverse square root of the low-rank matrix $\mathbf{A}^\top \mathbf{A}$ as $\mathbf{B} = (\mathbf{A}^\top \mathbf{A} + a\mathbf{I}_{n_{\text{odor}}})^{-1/2}$ for a small positive regularizing constant $a$. In our simulations, we set $a = 0.5$; we find empirically that our results are not substantially sensitive to small variations in $a$. Finally, we let $\boldsymbol{\Gamma} = \mathbf{B}\mathbf{Q} / \max\{|(\mathbf{A}\mathbf{B}\mathbf{Q})_{ij}|\}$.

In Fig. 4A, we smooth the concentration estimate timeseries using a 100 ms moving average. In Fig. 4B-C, we show cumulative estimates of the mean and variance. Concretely, given a concentration timeseries $\hat{c}_j(t)$, we estimate the mean and variance as

$$\mu_j(\tau) = \frac{\Delta t}{\tau} \sum_{t_{\text{low}} \leq t \leq t_{\text{low}} + \tau} \hat{c}_j(t) \tag{G.1}$$

and

$$\sigma_j^2(\tau) = \frac{\Delta t}{\tau} \sum_{t_{\text{low}} \leq t \leq t_{\text{low}} + \tau} \hat{c}_j(t)^2 - \mu_j(\tau)^2, \tag{G.2}$$

respectively, where we assume $\tau$ and $t$ are integer multiples of $\Delta t$. In Fig. 4D-E, we do the same except for times after $t_{\text{high}}$. In both cases, baselines were obtained by running the naïve sampling algorithm for $10^8$ steps with a burn-in period of $10^7$ steps. In Supp. Fig. G.3, we reproduce Fig. 4 showing estimates for individual odorants as well as the mean across odorants.

In Supp. Fig. G.4, we show a preliminary experiment with an alternative, mostly non-negative choice for $\boldsymbol{\Gamma}$. This gives the following three models:

- For the one-to-one code, we let $n_{\text{g}} = n_{\text{odor}}$, and simply set $\boldsymbol{\Gamma} = \mathbf{I}_{n_{\text{odor}}} / \max(A_{ij})$.
- For the naïvely distributed code, we let $n_{\text{g}} = 5n_{\text{odor}}$, and choose $\boldsymbol{\Gamma}$ as follows: We sample a sparse, non-negative random matrix $\mathbf{Q} \in \mathbb{R}^{n_{\text{odor}} \times n_{\text{g}}}$ with entries that are non-zero with density 0.15, and the non-zero entries are drawn uniformly on $[0, 1]$. Then, we define $\boldsymbol{\Gamma} = \mathbf{Q} / \max\{|(\mathbf{A}\mathbf{Q})_{ij}|\}$.
- For the geometry-aware code, we let $n_{\text{g}} = 5n_{\text{odor}}$, and choose $\boldsymbol{\Gamma}$ as follows: We sample a sparse, non-negative random matrix $\mathbf{Q} \in \mathbb{R}^{n_{\text{odor}} \times n_{\text{g}}}$ with entries that are non-zero with density 0.15, and the non-zero entries are drawn uniformly on $[0, 1]$. Then, we compute an approximate inverse square root of the low-rank matrix $\mathbf{A}^\top \mathbf{A}$ as $\mathbf{B} = (\mathbf{A}^\top \mathbf{A} + a\mathbf{I}_{n_{\text{odor}}})^{-1/2}$ for a small positive regularizing constant $a$. In our simulations, we set $a = 0.5$; we find empirically that our results are not substantially sensitive to small variations in $a$. Finally, we let $\boldsymbol{\Gamma} = \mathbf{B}\mathbf{Q} / \max\{|(\mathbf{A}\mathbf{B}\mathbf{Q})_{ij}|\}$.

This yields a naïvely distributed synaptic weight matrix $\mathbf{A}\boldsymbol{\Gamma}$ that is entirely non-negative, while the geometry-aware weight matrix has a small number of negative elements due to the fact that the inverse of a non-negative matrix need not be non-negative. We see that the behavior of this circuit is

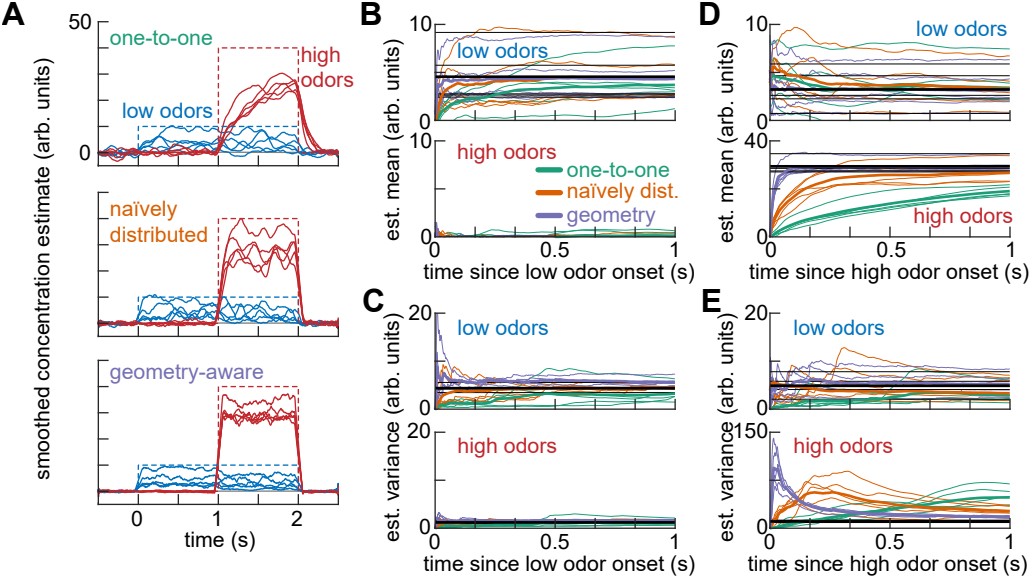

Figure G.3: Fast uncertainty estimation using Langevin sampling of the posterior, showing estimates for individual odorants. Here, we use a simple concentration estimation task in which 5 randomly-selected 'low' odorants out of a panel of 1000 appear at concentration 10 at time 0 s, and then a further 5 randomly-selected 'high' odorants appear at concentration 40 at time 1 s. **A**. Smoothed timeseries of instantaneous concentration estimates for low, high, and background odorants, for models with one-to-one (*top*), naïvely distributed (*middle*), and geometry-aware (*bottom*) codes. Background odorant estimates are shown as mean ± standard deviation over odorants. Dashed lines show true concentrations over time. **B**. Cumulative estimates of concentration mean for low (*top*) and high (*bottom*) odorants after the onset of the low odorants for one-to-one, naïvely distributed, and geometry-aware codes. Black lines indicate baseline estimates of the posterior mean. Thick colored lines indicate means over odorants, while thin lines show traces for individual odorants. **C**. As in **B**, but for the estimated variance. **D**. As in **B**, but after the onset of high odorants. **E**. As in **C**, but after the onset of high odorants. See Appendix G for details of our numerical methods.

similar to that observed in Fig. 4. As mentioned in the Discussion, an important topic for future work will be to devise a method to choose $\boldsymbol{\Gamma}$ that yields a fully non-negative, sparse synaptic weight matrix $\mathbf{A}\boldsymbol{\Gamma}$.

# H   Code and data availability

All code and data required to reproduce the figures presented is available under an MIT License at https://github.com/Pehlevan-Group/olfaction-geometry/.

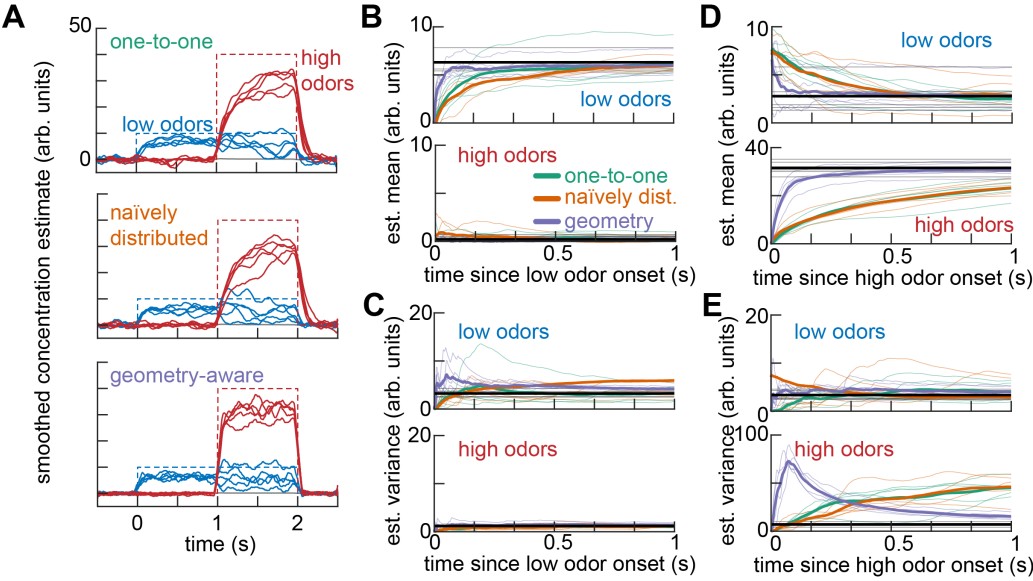

Figure G.4: Langevin sampling of the posterior using sparse non-negative distributed codes. Here, we use a simple concentration estimation task in which 5 randomly-selected 'low' odors out of a panel of 500 appear at concentration 10 at time 0 s, and then a further 5 randomly-selected 'high' odors appear at concentration 40 at time 1 s. This figure replicates Fig. 4, except that the code is distributed using a sparse non-negative random matrix rather than an orthogonal matrix. Smoothed timeseries of instantaneous concentration estimates for low, high, and background odors, for models with one-to-one (*top*), naïvely distributed (*middle*), and geometry-aware (*bottom*) codes. Background odor estimates are shown as mean ± standard deviation over odors. Dashed lines show true odor concentrations over time. **B**. Cumulative estimates of odor concentration mean for low (*top*) and high (*bottom*) odors after the onset of the low odors for one-to-one, naïvely distributed, and geometry-aware codes. Black lines indicate baseline estimates of the posterior mean. Thick lines indicate means over odors, and thin lines individual odors. **C**. As in **B**, but for the estimated variance. **D**. As in **B**, but after the onset of high odors. **E**. As in **C**, but after the onset of high odors. See Appendix G for details of our numerical methods.

