# OpenReview forum: "Neural Circuits for Fast Poisson Compressed Sensing in the Olfactory Bulb"
_NeurIPS.cc/2023/Conference — NeurIPS 2023 poster_

### Official Review · Reviewer_Qhvj · 2023-07-04

**Soundness:** 4 excellent
**Presentation:** 4 excellent
**Contribution:** 3 good
**Rating:** 6
**Confidence:** 4

**Summary:**

The authors derive a normative model to describe neural computations performed by the olfactory bulb. They describe in detail the biological interpretation of their model and show that the model makes realistic prediction about the biology of the olfactory bulb, matching the experimental observations. The authors validate the model on synthetic data, presenting different implementations of the model and evaluating its performance.

**Strengths:**

The paper is well written with excellent language and clear logic. The authors made a major effort in relating the equations they obtained to the biology of the network. The maths is rigorous and sound and is also well explained. The prediction of state-dependent inhibition obtained using the normative approach is elegant.

**Weaknesses:**

The bibliography is far too long and it makes it difficult for the reader to focus on references that are most relevant. While I like a well referenced paper, this is too much. I struggle to understand which papers I should read to evaluate better the present contribution.

Major - but hopefully addressable - concern. It is not fully clear whether the new model outperforms the existing ones. It is nice to see that the Poissonian model does better than the Gaussian one in predicting the effect of state-dependent inhibition. However, it would have been more relevant to compare the performances of the model with existing ones (Figures 3 and 4).

It is not easy to understand how far (or different) the model performance is from human performance in olfactory discrimination. Clarifying this would make it easier to evaluate the overall performance of the model.


**Questions:**

Can the authors quantify better the performance gain and or distill better the conceptual advance made by their study with respect to current computational models of olfactory discrimination? Maybe with direct application of competing models to the same simulated tasks? (this would be the key point  to increase my enthusiasm for the paper)

Can the authors distill the bibliography, only keeping the most relevant references?

How does the model perform in the fast identification of several odors, compared to previously existing (e.g. Gaussian CS) models?
Can the authors provide an estimate of the human performance in their task, as a benchmark for the model performance?
Based on Appendix G, it appears that naively distributed and the geometry-aware codes simulations ran using 5 times the number of granule cells compared to the one-to-one code. How much of the increase in model performance is due to the larger number of neurons and how much is due to a better coding scheme? Wouldn't it have been fairer to also set n_g = n_odor for the naively distributed and the geometry-aware codes? Additionally, from Fig. 3A it appears that the naively distributed code performance is quite low – also compared to the one-to-one code – for less than 10 odors. Authors should comment on why this is the case.



**Limitations:**

Yes, the authors describe well and fairly the  limitations in the Discussion section.

---

> ### Author Rebuttal · Authors · 2023-08-08
>
> We thank the referee for their careful reading of our work, and are glad to hear that they found it elegant, rigorous, and well-written. We reply to each of their concerns in turn below; we hope that the proposed changes strengthen their view that our paper should be presented at NeurIPS.
>
> ### Questions:
>
> 1. *Can the authors quantify better the performance gain and or distill better the conceptual advance made by their study with respect to current computational models of olfactory discrimination? Maybe with direct application of competing models to the same simulated tasks? (this would be the key point to increase my enthusiasm for the paper)*
>
> Please see our response to Point 3 below.
>
> 2. *Can the authors distill the bibliography, only keeping the most relevant references?*
>
> Thank you for this suggestion. We acknowledge that our paper is somewhat heavily referenced, which reflects the substantial body of work on the olfactory system. We will endeavor to distill each block of references down to the most relevant few.
>
> 3. *How does the model perform in the fast identification of several odors, compared to previously existing (e.g. Gaussian CS) models? Can the authors provide an estimate of the human performance in their task, as a benchmark for the model performance? Based on Appendix G, it appears that naively distributed and the geometry-aware codes simulations ran using 5 times the number of granule cells compared to the one-to-one code. How much of the increase in model performance is due to the larger number of neurons and how much is due to a better coding scheme? Wouldn't it have been fairer to also set n_g = n_odor for the naively distributed and the geometry-aware codes? Additionally, from Fig. 3A it appears that the naively distributed code performance is quite low – also compared to the one-to-one code – for less than 10 odors. Authors should comment on why this is the case.*
>
> Thank you for these questions. We respond to each of your points in turn:
> - Based on the extensive theoretical literature on compressed sensing in Poisson noise, it is difficult to make a meaningful, controlled comparison between a Gaussian CS model and a Poisson model if the data is generated using Poisson noise, as Gaussian algorithms will perform poorly even in the moderate-intensity regime. See, for example, Raginsky et al., “Compressed sensing performance bounds under Poisson noise,” IEEE Trans. Sig. Proc., 2010, and references therein. More biologically, it is not always clear how to make a fair comparison in terms of matching single-neuron time constants.
> - Second, the 1-to-1 code model is closely related to the Poisson CS model introduced by Grabska-Barwinska et al. 2013. Concretely, they differ only in that we split up the circuit to linearize the division, which allows an interpretation in terms of cell types in olfactory bulb circuits; the idea of performing gradient ascent on the log-posterior is otherwise identical. We alluded to this relationship in Line 113-114, but will add a sentence to further reify this point.
> - To the best of our knowledge, human psychophysics data for complex olfactory mixture tasks is not available. More precisely, we are not aware of experiments testing humans’ ability to detect single mixture components. Work from Liang and colleagues (Jinks, A. and Laing, D.G., 2001. The analysis of odor mixtures by humans: evidence for a configurational process. Physiology & behavior, 72(1-2), pp.51-63. or Laing, D.G. and Francis, G.W., 1989. The capacity of humans to identify odors in mixtures. Physiology & behavior, 46(5)) has investigated humans’ ability to identify all components of a mixture; humans perform poorly on this task.
> - It is of course true that the number of granule cells affects the speed of convergence and the accuracy of sampling. This is, however, precisely the reason we include the naively-distributed code, as it provides an intermediate point of comparison: comparing the naively-distributed code shows that the geometry-aware code’s performance gain relative to the 1-to-1 code is not due purely to the increased number of granule cells. We will elaborate on this point in our updated manuscript.

---

> > ### Comment · Reviewer_Qhvj · 2023-08-11
> >
> > Thanks to the authors for their detailed reply to this review and others. I do understand better why it is difficult to compare this model with others. Although I see this as a limitation, I also get the impression that the authors stretched the power of this approach to the limit and they did all what could and should have been done. I confirm my support for this study, and this is definitely a nice and elegant study, but I find it difficult to further raise to 7 the score because in the reply I could not appreciate (better than what I could appreciate from reading the submitting paper) the specific conceptual advance of this work with respect to the existing theories (normative or not) of olfactory coding. The reply to this question is shifted to the reply of point 3 which discusses more the issues arising when comparing with other work than in  distilling more powerfully the conceptual advance.

---

> > > ### Author Response · Authors · 2023-08-13
> > > **Response**
> > >
> > > We thank the reviewer for their kind words in response to our rebuttal, and are glad to hear that they found our paper “nice and elegant”. We would like to take this opportunity to further clarify what we see as the two most important conceptual advances of our work which makes in an important contribution to the field of olfaction:
> > >
> > > - We show that the geometry of inference is critical for fast inference in ~100ms. This has not been considered in previous models, which focused on unrealistic one-neuron-per-odor codes. This finding, in turn, shows the importance of considering geometry when trying to understand neural representations in the olfactory bulb (OB). Importantly, we show that it is the geometry in the space defined by the receptor affinity (or OSN activation) that crucially controls the speed of inference. This is a different view than other considerations of geometry in olfaction, who have often thought about the space of odorants (e.g. Zhou, Y., Smith, B.H. and Sharpee, T.O., 2018. Hyperbolic geometry of the olfactory space. Science advances, 4(8), p.eaaq1458.) or the space of subjective descriptors of odorants (e.g. Koulakov, A., Kolterman, B.E., Enikolopov, A. and Rinberg, D., 2011. In search of the structure of human olfactory space. Frontiers in systems neuroscience, 5, p.9271.). We think this is particularly important as unlike in other modalities such as vision or audition, where there are intuitive metrics along which to quantify the geometry of the sensory space, such as distance on the retina or tonotopy, similar metrics have been lacking in olfaction. We believe that thinking in terms of the geometry in the sensor space (OSN activations) will allow us to understand some of the transformation occurring in early olfactory processing.
> > >
> > > - We show that the specific circuit connectivity from our model can be mapped onto the general circuit motifs of the mammalian OB. The model also predicts observed experimental effects like state-dependent inhibition, offering a rationale for this architecture. Previous models did not achieve this level of interpretability.
> > >
> > > Taken together, these contributions constitute in our view a significant advance from past work, and an important step towards a computational understanding of the neural circuits implicated in early olfactory processing. In turn, we hope to motivate new experiments that aim to measure this geometry by measuring neural representations along large odorant panels.

---

> > > > ### Comment · Reviewer_Qhvj · 2023-08-14
> > > >
> > > > Thanks for these further clarification. My understanding of this reply is that a main conceptual novelty of the model is to argue that there is a structure in the geometry of activation in the OB, that they can derive this geometry intrinsically and largely unrelated to odor distances, and show that this geometry structure is important for fast odor inferences.
> > > > Is this correct?
> > > >
> > > > If that is the case, then a relevant study/approach would be Chong..Rinberg, Science 2020 (not cited), which reports an empirical derivation of distance metrics (and thus of geometry) of OB activity in perceptual discrimination space. Is there any contact between the present model and this older work? Is there any aspect of the current model that illuminates the older work, or the other way around?

---

> > > > > ### Author Response · Authors · 2023-08-14
> > > > >
> > > > > We thank the reviewer for their response, and for their continued patient engagement.
> > > > >
> > > > > Indeed, we argue that the geometry of representations in the OB is crucial to understand the computational process. In particular, a favorable geometry of the granule cell representation (and of the mitral/tufted-granule cell coupled dynamics, where we use the natural gradient to match the geometry of the statistical manifold) is needed to perform inference at behaviorally relevant speeds, due to the correlations in the OSN representation that result from broad receptor affinities. We also believe that the definition of an ‘odor distance’ is still an open question (as the extensive literature trying to propose different metrics suggests) and we argue that, perhaps, distance in the OSN responses space would be the appropriate metric.
> > > > >
> > > > > Chong et al., 2020 is an important paper as it introduces a technique to bypass the activation of receptors through natural odorants, offering much more flexibility in the stimulus design. However, in that paper the authors primarily look at the effect of changing the temporal structure of the stimulus, rather than the geometry as we consider it in our work.
> > > > >
> > > > > More related to our approach is a paper by the same group (Nakayama, H., Gerkin, R.C. and Rinberg, D., 2022. A behavioral paradigm for measuring perceptual distances in mice. Cell Reports Methods, 2(6).) that proposes behavioral measures of odor similarity. They do however only test pairs of single odors and binary mixtures, which makes the characterization of the space we consider (OSN activation is ~300 dimensions) possibly challenging. The approach could likely be generalized to allow for a more complete characterization of the geometry of the representation. It is, however, not entirely clear how to relate perceptual distances as measured through behavior to the geometry of OB responses, given the subsequent steps of cortical processing.
> > > > >
> > > > > The experimental methods used in these two studies could be used to test some of the ideas we put forward, as they would allow for a complete experimenter control of the geometry of neural activation. We will cite these papers in the Discussion section as part of an expanded discussion of experimental tests.

---

> > > > > > ### Comment · Reviewer_Qhvj · 2023-08-21
> > > > > >
> > > > > > Thanks, the plan to clarify the novelty in these terms and to refer to these relevant studies sounds reasonable. I hope to see the paper published. I have no objections to its publication.
> > > > > > For the Area Chairs, I leave a note that the the overall score of 6 is a lower bound to how I  value the paper. I do not feel that I should raise my score to 7, but I would raise it to 6.5 if intermediate scores were available.

---

### Official Review · Reviewer_cVJu · 2023-07-06

**Soundness:** 3 good
**Presentation:** 3 good
**Contribution:** 2 fair
**Rating:** 7
**Confidence:** 4

**Summary:**

This work presents a compressed sensing circuit model of the olfactory bulb. This circuit model performs the task of computing an estimate of odorant concentrations, given receptor activity. Notably, the proposed circuit takes great care to maintain biological plausibility. Linearized workarounds are found for non-linear circuit components. And the receptor activity is subject to Poisson noise, which is more biologically plausible than the typically considered Gaussian noise.

**Strengths:**

- The proposed circuit model is more biologically plausible than existing circuit models. And proposing a  circuit model in the first place avoids the weight transport problem of neural networks, or at least confines the problem to a regime where it could be plausibly explained.
- The proposed model recapitulates some biological phenomena that a model with a Gaussian noise assumption does not
- The proposed model performs better in the presence of distractors than the naive model
- Overall, the paper is well written, and seems like an improvement in explanatory power over existing work that the community follows

**Weaknesses:**

- Limitations are noted by the authors, namely the dimensionality of the odor space considered, but possible solutions, and an analysis of how the system would degrade are presented as well.
- There are simplifications, and points at which the model breaks with biology. For example, the roles of mitral and tufted cells are not distinguished. But for an understanding at the level of concepts, I believe this is alright.

**Questions:**

- Line 42: What does "distributed coding" mean in this context? Distributed across receptor types? Distributed across receptors?
- Could you briefly speculate what modifications might be made to the circuit to reproduce the nonlinear responses described in lines 84-85?
- Line 119: I'm missing something basic here. As far as I can tell, $n_g$ describes the number of receptor types and $n_{odor}$ is the number of odors. So, why is the case of $n_{g} > n_{odor}$ being considered? Isn't this at odds with the fact that there should be more odors than receptor types?
- Section 4.2: Do I understand correctly that the affinity matrix is fixed? Are there any pieces of the circuit that are "learned" or "fitted"?
- Could you help me understand why [Grabska-Barwinska et al. 2013](https://proceedings.neurips.cc/paper_files/paper/2013/file/2bcab9d935d219641434683dd9d18a03-Paper.pdf) refer to Poisson noise as an "unrealistic" assumption (page 8 under "future directions").

**Limitations:**

Yes, limitations are addressed, and initial solutions are sketched in the supplementary material, although I did not closely read these.

---

> ### Author Rebuttal · Authors · 2023-08-08
>
> We thank the reviewer for their careful assessment of our work and its limitations. We agree that building a model with distinguished roles for mitral and tufted cells will be an important objective for future studies. We respond to each of their questions below.
>
> # Questions:
>
> 1. *Line 42: What does "distributed coding" mean in this context? Distributed across receptor types? Distributed across receptors?*
>
> Thank you for this question. Here, “distributed coding” means that the presence and concentration of a specific odorant is not signaled by the activity of a specialized granule cell; rather, a response across many neurons in the population is triggered. We will expand our discussion of this point to avoid any loss of clarity.
>
> 2. *Could you briefly speculate what modifications might be made to the circuit to reproduce the nonlinear responses described in lines 84-85?*
>
> To model the nonlinear responses, a simple approach would be to apply a saturating nonlinearity to the linearly sensed concentrations. This is the approach that was taken by Qin, S., Li, Q., Tang, C. and Tu, Y., 2019. Optimal compressed sensing strategies for an array of nonlinear olfactory receptor neurons with and without spontaneous activity. Proceedings of the National Academy of Sciences, 116(41), pp.20286-20295.: if one assumes a simple competitive binding model for receptor activity, the resulting nonlinearity is $x/(1+x)$. We will add a sentence to the paper to comment on this point.
>
> 3. *Line 119: I'm missing something basic here. As far as I can tell, n_g describes the number of  receptor types and n_{odor} is the number of odors. So, why is the case of n_{g} > n_{odor} being considered? Isn't this at odds with the fact that there should be more odors than receptor types?*
>
> The number of receptor types in $n_{\mathrm{OSN}}$; $n_{\mathrm{g}}$ is the number of inhibitory granule cells. We keep $n_{\mathrm{OSN}} \ll n_{\mathrm{odor}}$ throughout, but allow $n_{\mathrm{g}} \geq n_{\mathrm{odor}}$. We will add a sentence to clarify that the number of receptor types is $n_{\mathrm{OSN}}$.
>
> 4. *Section 4.2: Do I understand correctly that the affinity matrix is fixed? Are there any pieces of the circuit that are "learned" or "fitted"?*
>
> Your understanding is correct: the affinity matrix is fixed. At the moment, the decoding matrix $\mathbf{\Gamma}$ is tuned by hand in the geometry-aware model to match the properties of the affinity matrix, such that $\mathbf{\Gamma} \mathbf{\Gamma}^{\top} \simeq (\mathbf{A}^{\top} \mathbf{A})^{+}$. We focus here on the problem of inference (and which can be studied by experimentally measuring the firing rates of neurons). There is no dynamic learning in the current version of our model. As noted in lines 312-323 of the Discussion, extending this model to incorporate synaptic plasticity will be an interesting topic for future work.
>
> 5. *Could you help me understand why Grabska-Barwinska et al. 2013 refer to Poisson noise as an "unrealistic" assumption (page 8 under "future directions").*
>
> Thank you for this question. For completeness, we reproduce the relevant sentences from Grabska-Barwinska et al:
>
> > We have made several unrealistic assumptions in this analysis. For instance, the generative model was very simple: we assumed that concentrations added linearly, that weights were binary (so that each odor activated a subset of the olfactory receptor neurons at a finite value, and did not activate the rest at all), and that noise was Poisson. None of these are likely to be exactly true.
>
> To the best of our understanding, the point being made here is that the true noise distribution is not perfectly Poisson. Of course, no choice of noise distribution is perfect, but Poisson noise is a better model for the statistics of OSN responses than Gaussian noise.

---

> ### Comment · Reviewer_cVJu · 2023-08-18
> **Response to authors**
>
> I thank the authors for the thorough response. I will continue to recommend acceptance.

---

### Official Review · Reviewer_DAo3 · 2023-07-07

**Soundness:** 3 good
**Presentation:** 4 excellent
**Contribution:** 3 good
**Rating:** 5
**Confidence:** 3

**Summary:**

The paper proposes a normative model for odor processing and representation in the olfactory bulb (of mammals). I did not track the conclusions, since I got distracted by the choices required to build a normative (mathematically tractable) model.

**Strengths:**

The paper is well written and the model is developed soundly (within the normative style) and with good detail.

If one accepts the sacrifices of biological detail needed to make the math clean, that is, if one supports the idea of normative modeling, the model may be a good one.


**Weaknesses:**

Note to Area Chair and authors:

I am not well suited to this paper, because I do not understand the motives behind the normative program, which prioritizes clean math over known biological information.  I see normative modeling as a Procrustes bed, and how far to distort biology for the sake of math is a matter that admits wide differences of opinion. I am uncomfortable with several of the choices (details below).

However, I realize that those sympathetic to the normative program will feel differently. Since they, and not I, are the audience for this paper, I'm happy to defer to their preferences.


**Questions:**

Context: I am not the right audience for this paper, since I do not grok the normative modeling approach. So the comments below can perhaps best be viewed not comments to be acted on, or even as criticisms, but as a window on how non-normative biological modelers might see the paper... for whatever that is worth, which may be not much :)

-----------

I don't get how this is a compressed sensing problem: the ORNs' chemical sensitivities correspond to our visual spectrum - they are simply the subset of chemicals that are measurable. Non-measurable chemicals effectively don't exist, and the sensing of detectable odors is not, to my understanding, random projection (see eg lines 171 - 172). I believe this is different from compressed sensing. I thought compressed sensing tied into olfaction further downstream, at the much larger (50x), sparsely firing layer (Mushroom Body in insects; I think there is a mammal equivalent), and in particular in the way the MB projects onto readout neurons (extrinsic neurons in insects) via random projections.(see Ganguli 2012).

For clarity, the biological interpretation/mapping (section 4) would go better much earlier, so the reader can map the mathematical elements to the OB as they show up. As it is, the math model is developed without context that would be highly useful.

I wonder if the model for odor concentrations (gamma prior with poisson noise at the ORNs) corresponds to actual odor plumes, or if it was chosen for mathematical convenience.

Equation 2: words to motivate what this equation is doing would be helpful (in general I prefer math motivated by words).

line 118: why is c (the odor) defined in terms of g (the projection neuron firing rates)? This seems backwards. Is it a convenience to make the math work, and if so is it stepping on any biological realities?

line 125: is it cricket to linearize the system? I thought the OB (AL) is strongly nonlinear. See for example Rachel Wilson's papers on divisive normalization in the AL. Here again (my refrain) I worry about biological sacrifices to make the math nice.

line 148: It sounds like g represent the lateral inhibition neurons. Is it the case that the mathematics accurately reflects what is concretely known about lateral inhibition in the OBL The symmetry of weights (expressed in the A-Gamma matrix) look highly unrealistic. Even having the projection neurons excite the inhibitory neurons is not accurate - both g and p are excited by the ORNs (caveat: my understanding is based in the insect AL, not the OB).

section 4.4: The inhibitory effects of the model may be reminiscent of experimental findings, but I believe those same findings can be more directly explained by known structure and dynamics of lateral inhibition within the OB. I would argue for subordinating the math to fit the known structure.

line 285: I don't see how negative synaptic weights can be justified (assuming that inhibition is expressed by a separate minus sign in the equations).

line 278- 279: I regret that I do not agree that the math clearly maps onto the OB

---

> ### Author Rebuttal · Authors · 2023-08-08
>
> We thank the reviewer for their careful and fair assessment of our manuscript, and appreciate the importance of their perspective given that we respectfully disagree about the fundamental utility of normative modeling. We emphasize that our normative model is not being used as a Procrustean bed, since we are not imposing exact conformity to a certain arbitrary standard. We are saying that if we posit certain reasonable axioms, non-trivial outcomes arise that agree with data. We are not saying anywhere that other explanations are not possible.  We respond to each of their concerns in turn below; we hope that this helps address some of their concerns regarding the biological relevance of our findings.
>
> 1. *I don't get how this is a compressed sensing problem [...]*
>
> In the OB (and across brain areas), compression and expansion occur at several stages of processing. In the mammalian olfactory system, there is indeed an expansion from the mitral/tufted cells to their projections in olfactory cortical areas (resembling the expansion of the projection into the MB in insects). But at the sensing stage, there is a compression from the space of possible odors to the space of OSN (and then M/T cell) responses that also maps onto a compressed sensing problem. Importantly, only a few out of the many detectable odorants are present in a given olfactory scene (see Krishnamurthy et al, 2022 Front. Comp. Neuro., or Koulakov and Rinberg 2011, Neuron)
>
> 2. *For clarity, the biological interpretation/mapping (section 4) would go better much earlier [...]*
>
> As with all of our paper, this organization is motivated by the normative framework - we aim to fully derive the mathematical model and only then connect it to biology. We appreciate that the reviewer will disagree with our preferences.
>
> 3. *I wonder if the model for odor concentrations (gamma prior with poisson noise at the ORNs) corresponds to actual odor plumes [...].*
>
> Modeling the odor concentration as gamma-distributed is a common approximation to the distribution of concentrations wafted on turbulent plumes, as it offers a useful balance of mathematical tractability with physical realism. Indeed, it can be physically motivated (see Celani, Villermaux, and Vergassola, 2014, PRX). A more realistic prior would include a spike at zero in addition to the approximately-Gamma slab of positive concentrations, but as we mention in the Discussion this is more challenging to sample and is left for future work. Similarly, the linear-mean OSNs with a Poisson noise model represents a compromise between detailed realism and mathematical tractability. As we mention below, a more detailed model would account for sensing nonlinearities due to receptor antagonism, but would be considerably more complicated. Therefore, these design choices reflect a balance between realism and conceptual simplicity. We will add a sentence to comment on the motivation for the Gamma prior to our updated manuscript.
>
> 4. *Words to motivate what this equation is doing would be helpful [...].*
>
> As noted in Line 111 above eq. (2), this is simply gradient ascent on the (logarithm of the) posterior probability resulting from (1). To make the motivation clearer, we propose to replace the sentence on Lines 109-110 starting with “Given this…” with “Given this likelihood and prior, we construct a neural circuit to compute the maximum a posteriori (MAP) estimate of the concentration $\mathbf{c}$ using gradient ascent on the log-posterior probability.”
>
> 5. *Why is c (the odor) defined in terms of g (the projection neuron firing rates)? [...]*
>
> It is not the underlying concentration of the odor that is defined in terms of the model granule cell firing rates, but rather the system’s estimate of the concentration. If the reviewer thinks it would be helpful to adopt notation to explicitly disambiguate the concentration estimate from the underlying concentration—perhaps by writing $\hat{\mathbf{c}}$ rather than just $\mathbf{c}$, we would be happy to do so.
>
> 6. *Is it cricket to linearize the system? [...]*
>
> As we noted in Lines 83-87, mammalian OSNs do indeed display nonlinear (saturating) responses when exposed to wide ranges of odor concentration. However, within a moderate concentration range as we use here, linear models provide an accurate description; see Gupta, Albeanu, and Bhalla, 2015 Nat Neurosci. 2015 (ref. [66]).
>
> 7. *[...] Is it the case that the mathematics accurately reflects what is concretely known about lateral inhibition in the OB?*
>
> In mammals, the granule cells receive excitatory input from OSNs only indirectly via the M/T cells, not directly (this is diagrammed in Figure 1A). This is an anatomical difference between OB and insect AL. Similarly, as we discussed in Lines 152-160, due to the fact that granule and M/T cells are coupled by dendrodendritic synapses, including a large number of reciprocal synapses, it is not implausible to assume approximately symmetric coupling.
>
> 8. *[...] I would argue for subordinating the math to fit the known structure.*
>
> We hope that the above clarification regarding the anatomy of the OB helps address some of your concerns regarding the biological plausibility of our model. We respectfully disagree with the reviewer that it is always beneficial to subordinate math to known biological structure; we would argue that depending on the level of understanding desired it can be useful to make simplifications for the sake of conceptual clarity.
>
> 9. *I don't see how negative synaptic weights can be justified [...]*
>
> As we discuss in the paragraph starting on Line 284, we agree that this feature is not entirely biologically satisfactory; our model is of course not perfect, but it is useful. In that paragraph, we offer potential accounts for a few negative weights, but this will be an important topic for future work.
>
> 10. *I regret that I do not agree that the math clearly maps onto the OB*
>
> We appreciate the reviewer’s reasons for disagreement.

---

> > ### Comment · Reviewer_DAo3 · 2023-08-12
> > **Re authors' responses**
> >
> > My thanks to the authors for thorough and thoughtful responses to questions. I'm happy to defer to other reviewers' opinions for reasons stated in my original review.
> > If convenient, I would still urge the authors to map the model to the OB as you go.
> > Re comparison to other models: To me this is less important than showing that a model with novel, plausible structural details can behave plausibly - this tells readers that these novel details may be worth looking into, ie it increases the menu.
> > Re references: A middle ground between lots and fewer references is to mark the key ones with asterisks, so there is a well-defined subset to concentrate on.

---

> > > ### Author Response · Authors · 2023-08-13
> > > **Response**
> > >
> > > We thank the reviewer for their response to our comments.
> > >
> > > As we pointed out, we also agree that direct comparison with other models might not be the most important point. But we believe that our model points out important aspects that have been overlooked previously. The geometry of the computation in the sensor space (OSN activation) should be measured and taken into account to understand the dynamics of computations in the olfactory bulb. In particular, the geometry of inference is critical for fast inference in ~100ms, which has not been considered in previous models that focused on unrealistic one-neuron-per-odor codes.. The architecture suggested by the Poisson model (and not present in the Gaussian one) does make predictions (such as state-dependent inhibition) that can be compared to experimental data.
> > >
> > > Regarding references, we will follow the reviewer’s recommendation and note which are the key references (either through an asterisk or by citing them first, followed by ‘see also …’ ).
> > >
> > > We will consider the reviewer’s suggested change in organization (map the model to OB as you go) when revising the manuscript for final submission. Although we will keep the overall structure of the paper the same, we will mention the mapping onto the circuit (pointing to the relevant part of Section 4) as we present the derivations in Section 3.

---

### Official Review · Reviewer_xx4u · 2023-07-07

**Soundness:** 3 good
**Presentation:** 4 excellent
**Contribution:** 2 fair
**Rating:** 6
**Confidence:** 1

**Summary:**

The authors develop a normative model for olfactory bulb using compressed sensing theory. They map the derived circuit model to known properties of cell types and  further hypothesize their roles.

**Strengths:**

I am not an expert in olfaction. However, I enjoyed reading this paper as it is very well written - introduces the basic neuroscience of olfaction, the state of the art and the contribution of the paper.

The rationale, and analyses seem correct, but I cannot comment on the novelty.

**Weaknesses:**

there is no connection to real data (except defining the distribution of A).

Since the goal of this paper is to build a normative circuit for olfaction based on biological principles, and since I am not an expert on olfaction, I cannot verify the design choices of the model.


**Questions:**

* Are the exact time-scales of dynamics biologically correct?


**Limitations:**

Limitations have been appropriately discussed.

---

> ### Author Rebuttal · Authors · 2023-08-08
>
> We thank the reviewer for their careful assessment of our work, and are pleased to hear that they found it enjoyable to read. We’d like to note that beyond using OSN activity measurement to define the affinity matrix $A$, we also reproduce the state-dependent inhibition effect observed in Arevian et al., Nature Neurosci, 2008, and this effect does not occur in a circuit model derived for Gaussian noise in the OSNs. Moreover, we tune the timescales of the neural dynamics to match experimentally measured membrane time constants of mitral/tufted and granule cells (see below).
>
> ### Questions:
>
> *Are the exact time-scales of dynamics biologically correct?*
>
> Yes, the values for the membrane time constant of the neurons were chosen to match values measured experimentally and published in the literature. For mitral/tufted cells we used $\tau_p = 20ms$ (see Burton, S.D. and Urban, N.N., 2014. Greater excitability and firing irregularity of tufted cells underlies distinct afferent‐evoked activity of olfactory bulb mitral and tufted cells. The Journal of Physiology, 592(10), pp.2097-2118.) For granule cells, we used $\tau_g = 30 ms$ (see Burton, S.D. and Urban, N.N., 2015. Rapid feedforward inhibition and asynchronous excitation regulate granule cell activity in the mammalian main olfactory bulb. Journal of Neuroscience, 35(42), pp.14103-14122.) We will make this point more clearly and cite the relevant papers in the updated version of the manuscript.

---

### Official Review · Reviewer_Wphi · 2023-07-20

**Soundness:** 4 excellent
**Presentation:** 4 excellent
**Contribution:** 3 good
**Rating:** 7
**Confidence:** 3

**Summary:**

The study introduces a novel compressed sensing (CS) rate-based circuit model
relying on Poisson noise models instead of Gaussian noise models. A central
claim is that the model can be mapped onto the neural circuits of the olfactory
bulb while considering neurophysiological features and a biologically plausible
connectivity scheme. Importantly, compared to previous Gaussian CS models, the
Poisson noise model enables distributed odor coding instead of indicating odor
presence via the activity of a single neuron.

**Strengths:**

**Originality:** The authors propose a novel extension of prior work on
compressed sensing capturing properties of the the olfactory bulb using a
Poisson noise model. They convincingly show that their model can explain
observations found in in vitro recordings while outperforming previous CS
models.

**Quality:** The experiments are well designed and performed convincingly. The
interpretations and intuitions in this work are carefully constructed and
eloquently presented. The figures complement the main text well. The discussion
section is one of the highlights of the paper.

**Clarity:** Exceptionally clear writing and presentation. Sections are
structured well and link together to form a coherent picture.

**Significance:** Excellent starting poiont for spike-based implementations of
olfactory microcircuits and how the framework of compressed sensing can be used
to deduce the functionality of circuits under investigation.

**Weaknesses:**

Minor remarks:

1. Panels B to E in Figure 4 appear crammed, chaotic and could be plotted
   differently. Maybe figures with individual odorants could be provided in the
   supplement instead and the main figures just show the thicker lines?
   Alternatively, the transparency of the lines could be reduced?
2. Figure 1C is unreadable after printing on paper. Even on screen, the non-zero
   entries are hard to spot.

**Questions:**

1. What are the simulation parameters for the capacity simulations and the
   sampling simulations? Some parameter settings seem to be missing in the
   supplement.
2. If in-vitro recordings are available, wouldn't it be possible to estimate
   more model parameters from data directly?
3. What do you want to show using Figure 1C that could not be shown using
   summary statistics?


**Limitations:**

Limitations of the model are adequately addressed in the discussion section.

---

> ### Author Rebuttal · Authors · 2023-08-08
>
> We thank the referee for their positive assessment of our work. We hope that our revisions will resolve their concerns about the clarity of the figures and of simulation parameters.
>
> ### Weaknesses:
>
> Minor remarks:
>
> 1. *Panels B to E in Figure 4 appear crammed, chaotic and could be plotted differently. Maybe figures with individual odorants could be provided in the supplement instead and the main figures just show the thicker lines? Alternatively, the transparency of the lines could be reduced?*
>
> Thank you for this comment; we will follow your suggestion of providing a figure with individual odorants only in the supplement. We include the updated main-text figure showing only the thicker lines representing the averages in the PDF attached to the common response.
>
> 2. *Figure 1C is unreadable after printing on paper. Even on screen, the non-zero entries are hard to spot.*
>
> Thank you for noticing this legibility issue; after printing we agree that it is difficult to spot the non-zero entries. As we do not want to show much about the resulting sensing matrix other than the distribution of individual entries and the fact that it will have a few large entries, which Figure 1B is adequate to show (c.f. your 3rd question), we will remove it from the paper.
>
> ### Questions:
> 1. *What are the simulation parameters for the capacity simulations and the sampling simulations? Some parameter settings seem to be missing in the supplement.*
>
> Thank you for noticing this omission. For the capacity and sampling simulations, the time constants used were $\tau_{p} = 0.020\ \mathrm{s}$ and $\tau_{g} = 0.030\ \mathrm{s}$, matching the state-dependent inhibition simulations and experimentally measured values for mitral/tufted and granule cells respectively (see below). We set the baseline rate to $r_{0} = 1$, and the prior strength to be $\lambda = 1$; based on some experimentation our results appear relatively insensitive to small variations in these choices. We will make sure to state these parameter settings more clearly in the updated manuscript.
>
> 2. *If in-vitro recordings are available, wouldn't it be possible to estimate more model parameters from data directly?*
>
> Thank you for this suggestion. We already chose the time constants to match experimental measurements. For mitral/tufted cells we used $\tau_p = 20ms$ (see Burton, S.D. and Urban, N.N., 2014. Greater excitability and firing irregularity of tufted cells underlies distinct afferent‐evoked activity of olfactory bulb mitral and tufted cells. The Journal of Physiology, 592(10), pp.2097-2118.) For granule cells, we used $\tau_g = 30 ms$ (see Burton, S.D. and Urban, N.N., 2015. Rapid feedforward inhibition and asynchronous excitation regulate granule cell activity in the mammalian main olfactory bulb. Journal of Neuroscience, 35(42), pp.14103-14122.) We will clearly mention this in the updated version of our manuscript. We agree that it would be interesting to constrain the values of other parameters based on experiment; we will comment on this in our revised manuscript.
>
> 3. *What do you want to show using Figure 1C that could not be shown using summary statistics?*
>
> As noted above, we will remove Figure 1C, as the distributions shown in Figure 1B are sufficient to communicate our point.

---

> > ### Comment · Reviewer_Wphi · 2023-08-21
> >
> > I thank the authors for their responses and proposed changes to the paper. After reading the other reviews, I think that this submission is suitable for publication.

---

### Author Rebuttal · Authors · 2023-08-08

We thank the reviewers for their favorable comments on our manuscript. We outline the primary changes we propose to make to the manuscript here, and respond to the reviewer’s detailed comments point-by-point individually. The largest changes to our manuscript will be as follows:
- Following Reviewer Wphi’s suggestion, we will simplify Figure 4 by removing the lines for individual odors, and include a version with those lines only in the supplement. We include this revised figure in the PDF attachment.
- We will remove Figure 1C, which Reviewer Wphi found to be illegible.
- We will add a new figure showing how the capacity of the different algorithms tested in Figure 3 scales with the number of odors. This panel is included in the PDF.
- We will elaborate on the motivation underlying our choice of a Gamma prior, as requested by Reviewer DAo3.
- We will mention that we chose the model membrane time constants to match experimental measurements, addressing a question raised by Reviewers Wphi and xx4u.

---

### Decision · Program_Chairs · 2023-09-21

**Decision:**

Accept (poster)

**Comment:**

A new model for olfaction which performs compress sensing in a biologically plausible way with an extension for Bayesian inference to measure uncertainty. The tools developed here could see application to other domains where uncertainty is important.

There is great interest in the community in biologically plausible ML. This is a fairly unique arena in which to consider biological plausibility.

Multiple reviewers noted that contextualizing this work better would be beneficial. A large part of the value of this publication in NeurIPS is explaining the problems in this domain clearly where members of the community can engage with both the problem and the models offered here. As came up in the rebuttals, a precise discussion about what aspects are or are not biologically plausible, what problems are being solved and what problems are left, would both be important for a final submission. The more clearly the authors articulate this, the more engagement with their work there will be.